# The impact of biogenic, anthropogenic and biomass burning volatile organic compound emissions on regional and seasonal variations in secondary organic aerosol

Jamie. M. Kelly[1], Ruth M. Doherty[1], Fiona. M. O'Connor[2], and Graham W. Mann[3]

[1]School of GeoSciences, The University of Edinburgh, U.K
[2]Met Office, Hadley Centre, Exeter, U.K
[3]National Centre for Atmospheric Science, School of Earth and Environment, University of Leeds, Leeds, U.K

*Correspondence to*: Jamie Kelly (j.kelly-16@sms.ed.ac.uk)

**Abstract.** The global secondary organic aerosol (SOA) budget is highly uncertain, with global annual SOA production rates, estimated from global models, ranging over an order of magnitude and simulated SOA concentrations underestimated compared to observations. In this study, we use a global composition-climate model (UKCA) with interactive chemistry and aerosol microphysics to provide an in-depth analysis of the impact of each VOC source on the global SOA budget and its seasonality. We further quantify the role of each source on SOA spatial distributions, and evaluate simulated seasonal SOA concentrations against a comprehensive set of observations. The annual global SOA production rates from monoterpene, isoprene, biomass burning and anthropogenic precursor sources is 19.9 19.6, 9.5 and 24.6 Tg (SOA) a$^{-1}$ respectively. When all sources are included, the SOA production rate from all sources is 73.6 Tg (SOA) a$^{-1}$, which lies within the range of estimates from previous modelling studies. SOA production rates and SOA burdens from biogenic and biomass burning SOA sources peak during northern hemisphere (NH) summer. In contrast, the anthropogenic SOA production rate is fairly constant all year round. However, the global anthropogenic SOA burden does have a seasonal cycle which is lowest during NH summer, which is probably due to enhanced wet removal. Inclusion of the new SOA sources also accelerates the ageing by condensation of primary organic aerosol (POA), making it more hydrophilic, leading to a reduction in the POA lifetime. With monoterpene as the only source of SOA, simulated SOA and total organic aerosol (OA) concentrations are underestimated by the model when compared to surface and aircraft measurements. Model agreement with observations improves with all new sources added, primarily due to the inclusion of the anthropogenic source of SOA, although a negative bias remains. A further sensitivity simulation was performed with an increased anthropogenic SOA reaction yield, corresponding to an annual global SOA production rate of 70.0 Tg (SOA) a$^{-1}$. Whilst simulated SOA concentrations improved relative to observations, they were still underestimated in urban environments and overestimated further downwind and in remote environments respectively. On the other hand, the inclusion of SOA from isoprene and biomass burning did not improve model–observations biases substantially except at one out of two tropical locations. However, these findings may reflect the very limited availability of observations to evaluate the model, which are primarily located in the NH mid-latitudes where anthropogenic emissions are high. Our results highlight that, within the current uncertainty limits in SOA sources and reaction yields, over the NH mid-latitudes, a large

anthropogenic SOA source results in good agreement with observations. However, more observations are needed to establish the importance of biomass burning and biogenic sources of SOA in model agreement with observations.

## 1 Introduction

Organic Aerosol (OA) is important from both air quality and climate perspectives. Measurements across the northern hemisphere (NH) mid-latitudes suggest OA represents between 18 and 70 % of fine aerosol mass depending on location and atmospheric conditions (Zhang et al., 2007). However, due to coarse grid resolutions and uncertainties in the SOA lifecycle, global chemistry transport models and general circulation model systematically underpredict observed OA concentrations in both urban (mean normalised bias = -62 %) and remote (mean normalised bias = -15 %) environments (Tsigaridis et al., 2014). Therefore, assessment of subsequent OA impacts on air quality (Hodzic et al., 2016) and climate (Scott et al., 2015) suffer from a lack of confidence.

OA can be emitted as primary organic aerosol (POA) or formed as secondary organic aerosol (SOA). SOA is formed from the oxidation products of volatile organic compounds (VOCs) and semi volatile and intermediate volatility organic compounds (S/IVOCs). Across the NH mid-latitudes, SOA accounts for 64, 83 and 95 % of observed surface OA submicron mass in urban, urban downwind and remote environments respectively (Zhang et al., 2007). Despite the observed ubiquity of SOA in the atmosphere, models generally perform poorly at reproducing observed SOA concentrations (Tsigaridis et al., 2014). Using either global (Hodzic et al., 2016) or regional (Chen et al., 2009) scale models, relatively good agreement between simulated and observed SOA concentrations in remote environments has been found. In contrast, in urban environments, simulated SOA concentrations are substantially lower than observed (Hodzic et al., 2009; Heald et al., 2011; Khan et al., 2017).

Estimated annual global SOA production rates, based on bottom-up approaches using global models, range from 12 to 480 Tg (SOA) a$^{-1}$ (Kanakidou et al., 2005;Heald et al., 2011;Tsigaridis et al., 2014;Hodzic et al., 2016;Shrivastava et al., 2015). This large inter-model spread in simulated SOA production rates is likely due to an incomplete understanding of the different sources of SOA and a lack of observations available to constrain models. Global annual SOA production rates, estimated from top-down methods, such as scaling of the sulphate budget (Goldstein and Galbally, 2007), or constraining the SOA budget using satellite data (Heald et al., 2010) or in-situ observations (Spracklen et al., 2011), are substantially greater and even more uncertain (range 50 – 1820 Tg (SOA) a$^{-1}$). Uncertainty ranges for the global POA budget are narrower, with estimates of global POA production ranging from 34 to 144 Tg (POA) a$^{-1}$ (Tsigaridis et al., 2014).

Emitted from both natural and anthropogenic sources, SOA has important climatic impacts. SOA can perturb the energy balance of the Earth by absorption and scattering (aerosol-radiation interactions) and by altering cloud properties (aerosol-cloud interactions) (Forster and Ramaswamy, 2007). However, climatic impacts of SOA are very sensitive to the SOA budget. For example, in a multi-model study, estimates of the increase in the global SOA burden since preindustrial times ranged from 0.09 to 0.97 Tg (SOA), which resulted in a direct radiative effect ranging from -0.21 to -0.01 W m$^{-2}$ (Myhre et al., 2013). As an aerosol with a large natural source, SOA also contributes to the uncertainty in preindustrial aerosol

concentrations, which is the dominant contribution to the uncertainty in the aerosol cloud albedo effect forcing (Carslaw et al., 2013).  Hence, reducing uncertainty in the SOA budget is crucial for constraining radiative forcing estimates.

The identity of SOA precursors is critical to our understanding of the SOA budget and its lifecycle, however, the dominant precursors of SOA remain highly uncertain in their magnitudes and spatial distributions. Biogenic volatile organic compounds (bVOCs) are considered important sources of SOA due to high emissions (Guenther et al., 2012) and fast reactivity. Laboratory and field studies suggest that monoterpene ($C_{10}H_{16}$) (Yu et al., 1999), isoprene ($C_5H_8$) (Ng et al., 2008), sesquiterpenes ($C_{15}H_{24}$) (Tasoglou and Pandis, 2015) and the heterogeneous uptake of glyoxal (Volkamer et al., 2007) are all important biogenic sources of SOA. Global biogenic SOA production rates, estimated using global models, range between 2.86 and 97.5 Tg (SOA) a$^{-1}$ (Henze et al., 2008;Heald et al., 2008;Farina et al., 2010;Hodzic et al., 2016). Isoprene (Bonsang et al., 1992) and monoterpene (Yassaa et al., 2008) are also emitted from phytoplankton. Although the SOA production rates from marine VOCs is small, estimated at 5 Tg (SOA) a$^{-1}$ (Myriokefalitakis et al., 2010), marine SOA may be important from a climate perspective due to changes to cloud-condensation nuclei (Meskhidze et al., 2011).  Many global models only consider biogenic sources in their formation of SOA (Tsigaridis et al., 2014); other studies that include a number of different precursor types suggest that biogenic sources contribute 74 % (Hodzic et al., 2016)  to 95 % (Farina et al., 2010) to the annual global total SOA production rate.

According to laboratory studies, anthropogenic VOCs (e.g. aromatics) yield only a small amount of SOA (Odum et al., 1997). Therefore, the inclusion of anthropogenic SOA in global models based on such modest yields produces little SOA production (1.6 – 3.1 Tg (SOA) a$^{-1}$) and almost negligible SOA concentrations (Farina et al., 2010;Heald et al., 2011). However, over the NH mid-latitudes, observed SOA concentrations are highest in urban environments (Zhang et al., 2007). In a top-down approach, where biogenic, biomass burning and anthropogenic SOA sources were scaled, Spracklen et al. (2011) found that their simulated model bias was minimised when the global SOA budget was dominated by anthropogenic precursors, with an anthropogenic SOA production rate of ~100 Tg (SOA) a$^{-1}$ and a biogenic SOA production rate of 13 Tg (SOA) a$^{-1}$. The magnitude of anthropogenic SOA production as well as dominance over biogenic SOA production estimated from this top-down study is in stark contrast to bottom up estimates from global modelling studies. However, it remains unclear whether the anthropogenic dominance of global SOA production found in Spracklen et al. (2011) simply reflects the location of observations used to constrain this estimate; observations were primarily located in the NH mid-latitudes where anthropogenic emissions are highest. Additionally, the reaction yield required to reach a production rate of 100 Tg (SOA) a$^{-1}$ of anthropogenic SOA exceeds the reaction yield derived from laboratory studies (Odum et al., 1997). Furthermore, a high SOA production rate from anthropogenic sources produced positive models biases compared to observed SOA concentrations in remote environments (Spracklen et al., 2011) and at higher altitudes (Heald et al., 2011).

Laboratory (Grieshop et al., 2009) and field campaigns (Cubison et al., 2011) suggest biomass burning VOCs and S/IVOCs can form SOA. Using observations, Cubison et al. (2011) estimated the global SOA production rate from biomass burning at 1 – 15 Tg (SOA) a$^{-1}$. This is consistent with Spracklen et al. (2011), who estimated a global SOA production rate from biomass burning of 3 – 26 Tg (SOA) a$^{-1}$ using a top-down approach. S/IVOCs are estimated to account for between 15 –

37 % of biomass burning carbonaceous emissions (Stockwell et al., 2015;Yokelson et al., 2013). Due to the limited knowledge of S/IVOCs emissions and chemistry, assumptions are required when implementing biomass burning S/IVOCs into global models. As a consequence biomass burning SOA production rates, estimated using global models, range substantially from 15.5 Tg (SOA) a$^{-1}$ (Hodzic et al., 2016) to 44 – 95 Tg (SOA) a$^{-1}$ (Shrivastava et al., 2015). Therefore, biomass burning remains an additional highly uncertain and potentially important source of SOA. Therefore, biomass burning remains an additional highly uncertain and potentially important source of SOA.

Anthropogenic and biomass burning emissions include S/IVOCs, which in addition to VOCs, can also contribute to SOA formation (Robinson et al., 2007). As traditional emissions inventories, such as van der Werf et al. (2010), account for only a fraction of the emitted S/IVOCs, most modelling studies completely neglect the role of S/IVOCs as sources of SOA. Alternatively, SOA formation from S/IVOCs can be artificially accounted for by scaling up reaction yields of VOCs. Assumptions in the amount of S/IVOCs missing from traditional emissions inventories introduces uncertainty, with estimates ranging from 0.25 – 2.8 times POA emissions (Robinson et al., 2010;Shrivastava et al., 2008). Estimates of global annual-total S/IVOC emissions range from 54 Hodzic et al. (2016) to 450 Tg a$^{-1}$ (Shrivastava et al., 2015). Despite these uncertainties, S/IVOCs have been included in a number of models. Both Hodzic et al. (2016) and Tsimpidi et al. (2016) agree that the sum of the global SOA burden from anthropogenic and biomass burning precursors is dominated by S/IVOCs (72 and 69 % respectively).

The heterogeneous production of SOA may be an additionally important source of SOA, which is often not included in models (e.g., see (Tsigaridis et al., 2014). Soluble and polar organic vapours, which are too volatile to condense, may be taken up by the aqueous phase, either in aerosol liquid water or cloud liquid water. Within the aqueous phase, oxidation may lead to less volatile products which, after liquid evaporation, remain in the aerosol phase (Ervens, 2015). One example of a polar water soluble compound, which is too volatile to directly condense, but may form SOA within the aqueous phase, is glyoxal (Volkamer et al., 2007). Recent estimates of the global annual-total SOA production rate within the cloud and aerosol phases are 13 – 47 and 0 – 13 Tg (SOA) a$^{-1}$ respectively (Lin et al., 2014;Lin et al., 2012). Aqueous production may be an important source of SOA, but several uncertainties remain, including the amount of cloud and liquid water in the atmosphere, and how to simulate the uptake of organic gases onto aqueous surfaces.

The diversity in treatment of SOA formation within global models was highlighted in a recent multi-model study (Tsigaridis et al., 2014). Of the 31 models included in this study, 11 models treat SOA formation by directly 'emitting' SOA from vegetation. In doing so, these schemes neglect the role of many important atmospheric conditions on SOA formation rates. For example, the effect of oxidant levels on both reaction rates and yields of SOA precursors are neglected. Some models simulate SOA precursor emissions and subsequent oxidation to lower volatility compounds, which then condense to form SOA. In such SOA schemes, the effect of oxidant levels on SOA precursor reaction kinetics are accounted for. However, in studies where a single reaction yield is assumed throughout the atmosphere (Chung and Seinfeld, 2002;Tsigaridis and Kanakidou, 2003;Spracklen et al., 2011), the effects of atmospheric conditions on reaction yields are neglected. However, chamber experiments show that SOA yields vary substantially with different atmospheric environmental conditions for each

of the main SOA precursor species: reaction yields of monoterpenes (Sarrafzadeh et al., 2016), isoprene (Rattanavaraha et al., 2016) and aromatics (Ng et al., 2007b) are all affected by oxidant levels.

. Environmental chambers are typically used to elucidate hydrocarbon oxidation mechanisms and SOA yields. Several recent advances in SOA environmental chamber studies have been made, including the influence of NOx and humidity on SOA yields. Early estimates of SOA yields from aromatic compounds, which were conducted in relatively high nitrogen oxide ($NO_x$ = NO and $NO_2$) concentrations, range between 5 – 10 % (Odum et al., 1997;Odum et al., 1996). However, more recent studies have shown that SOA yields from aromatic compounds are stongly influenced by NOx, which has therefore motivated the re-evaluation of laboratory deriverd SOA yields. Ng et al. (2007b) also observed an SOA yield from aromatic VOCs of 5 – 10 % under high-$NO_X$ conditions. However, under low-$NO_X$ conditions, Ng et al. (2007b) measured substantially higher SOA yields of 37, 30 and 36 % for benzene ($C_6H_6$), toluene ($C_7H_8$)  and xylene ($C_8H_{10}$), respectively. Also under low-$NO_X$ conditions, Chan et al. (2009) observed an SOA yield of 73 % from naphthalene ($C_{10}H_8$). Environmental chamber studies have also found similar relationships between NOx and SOA yields for several other important species, including monoterpene (Ng et al., 2007a) and isoprene. In addition, water vapour may also affect SOA yields. For aromatic compounds, both positive (White et al., 2014) and negative (Cocker et al., 2001) correlations between aromatic SOA yields and relative humidity have been observed in chamber studies, hence, the role of water vapour in aromatic oxidation is not yet clear. These environmental chamber studies have helped elucidate oxidation mechanisms, and linked them to the strength of SOA production (i.e. yields). However, global models commonly use a single SOA yield throughout the atmosphere. Therefore, the demonstration of variable SOA yields in laboratory studies results in greater difficulty in selecting an SOA yield for a global model. Also, within an environmental chamber, uptake of organic compounds by the surface walls (known as 'wall losses') can occur. Traditionally, this process was assumed to be non-negligible, resulting in the potential for environmental chamber studies to underestimate the reaction yield. Zhang et al. (2014) found that reaction yields of toluene, an anthropogenic source of SOA, were under estimated by a factor of four due to wall losses during chamber studies. This has a significant effect on simulated SOA production. Updating reaction yields to account for wall losses in global models resulted in an increase in the global annual biogenic SOA production rate from 21.5 to 97.5 Tg (SOA) $a^{-1}$ (Hodzic et al., 2016).

Volatility is another important aspect of OA. Whereas oxidation products from biogenic VOCs are predominantly compounds with extremely low volatility, so-called ELVOCs, (e.g. Ehn et al., 2014), there is considerable evidence that anthropogenic SOA is primarily composed of semi-volatile material. A large component of POA is also semi-volatile. For example, combustion POA, either generated from gasoline (May et al., 2013b) or diesel (May et al., 2013c) within laboratory studies suggests POA is primarily semi-volatile (i.e. only a small proportion consists of ELVOCs). Also, analysis of POA emitted from the burning of vegetation within laboratory studies suggests POA is semi-volatile (May et al., 2013a). Furthermore, analysis of OA, generated in both laboratories (Huffman et al., 2009b) and field campaigns (Huffman et al., 2009a) suggests OA is semi-volatile. However, some specific cases suggest a proportion of OA is non-volatile (Cappa and Jimenez, 2010;Jimenez et al., 2009;Vaden et al., 2011). Considering laboratory and field studies suggest OA is dominated by the semi-volatile fraction, the treatment of volatility of OA within global models ranges

substantially. Of the 31 state-of-the-art global models included in AeroCom, only 12 treat OA as semi-volatile (Tsigaridis et al., 2014). Recently, the effects of volatility on SOA were quantified in a global modelling study by Shrivastava et al. (2015). These authors estimate that the global annual-total SOA production rate varies by almost a factor of 2 depending on whether OA is treated as semi-volatile or non-volatile. The particle-phase state may be another important factor but is poorly

characterised (Shiraiwa et al., 2017), and is dependent on relative humidity and SOA precursor (Hinks et al., 2016;Bateman et al., 2016)" . Overall, global models show a clear systematic low bias in simulated SOA concentrations in comparison to observations (Tsigaridis et al., 2014). Current global models represent a broad spectrum in chemical complexity, ranging from essentially no chemical-dependence on SOA production up to moderate-complexity SOA mechanisms with volatility basis set (VBS) (Donahue et al., 2006). Whether this systematic underprediction of SOA is due to incomplete SOA sources in the

models or underprediction SOA yields is not yet clear. Although numerous studies have investigated SOA production in global models, these have mostly focussed on individual sources, with an aim to reduce model biases. Very few studies include all major sources of SOA. Some studies suggest that the global SOA budget is dominated by biogenic sources (Hodzic et al., 2016), whereas others suggest it is dominated by biomass burning (Shrivastava et al., 2015). Additionally, Spracklen et al. (2011) indicates that biogenic SOA can be "anthropogenically enhanced" in polluted regions.

In this study, a global chemistry and aerosol model (UKCA) is used to simulate SOA concentrations from all the VOC emission source types described above: monoterpene, isoprene, anthropogenic and biomass burning. Other mechanisms of SOA production, such as S/IVOCs and heterogeneous production, may also be important, but in this study, the focus is on VOCs. The novelty of this study is that a global model is used to simulate SOA and POA from a number of different sources, evaluating simulated concentrations against a consistent set of observations to provide new insights into the seasonal influence

of these different SOA precursor sources. The paper is organised as follows. Section 2 describes the modelling approach and the measurements used to evaluate the model are described in Section 3.  Results on the global SOA production rates, spatial POA and SOA distributions, and comparisons with surface and aircraft observations are explored in Section 4, with discussion and concluding remarks in Section 5.

## 2. Chemistry-climate model description

For this study, we use the United Kingdom Chemistry and Aerosol (UKCA) model (Morgenstern et al., 2009; Mann et al., 2010; O'Connor et al. 2014) coupled to the  Global Atmosphere 4.0 (GA4.0) configuration (Walters et al., 2014) of the Hadley Centre Global Environmental Model (Hewitt et al., 2011) version 3 (HadGEM3). We use an atmosphere-only configuration with prescribed sea surface temperature and sea ice fields based on 1995-2004 reanalyses data (Reynolds et al., 2007). The model is run at a horizontal resolution of N96 (1.875° longitude by 1.25° latitude).  The vertical dimension has 85 terrain-

following hybrid-height levels distributed from the surface to 85 km. Horizontal winds and temperature in the model are nudged towards ERA-Interim reanalyses (Dee et al., 2011)  using a Newtonian relaxation technique  with a relaxation time constant of 6 hours (Telford et al., 2008). There is no feedback from the chemistry or aerosols onto the dynamics of the model;

this ensures identical meteorology across all simulations, so that differences in modelled SOA concentration are solely due to differences in SOA sources.

The United Kingdom Chemistry and Aerosol (UKCA) model  used in this study combines the "TropIsop" tropospheric chemistry scheme from O'Connor et al. (2014) with the stratospheric chemistry scheme from Morgenstern et al. (2009). There are 75 species with 285 reactions. This includes odd oxygen ($O_x$), nitrogen ($NO_y$), hydrogen ($HO_x$ = OH + $HO_2$), and carbon monoxide (CO). Hydrocarbons included are methane, ethane, propane, isoprene and monoterpene. Isoprene oxidation follows the Mainz Isoprene Mechanism (Poschl et al., 2000) which is described in detail in O'Connor et al. (2014).

The aerosol component of UKCA is the 2-moment modal version of the Global Model of Aerosol Processes (GLOMAP-mode) (Mann et al., 2010). Aerosol components considered are sulphate ($SO_4$), sea salt (SS), black carbon (BC), primary organic aerosol (POA) and secondary organic aerosol (SOA). Both aerosol mass and number are transported in seven internally mixed log-normal modes (four soluble and three insoluble). Aerosol growth occurs via nucleation, coagulation, condensation, ageing, hygroscopic growth and cloud processing. Condensation ageing transfers hydrophobic particles into the hydrophilic mode when a 10 monolayer is formed on the surface a hydrophobic particle. Dry deposition and gravitational settling of aerosol follows Slinn (1982) and  Zhang et al. (2012), respectively. Grid-scale wet deposition of aerosol occurs via nucleation scavenging and impact scavenging. Subgrid-scale wet deposition occurs via plume scavenging (Kipling et al., 2013). New particle formation by binary homogenous nucleation of sulphuric acid ($H_2SO_4$) follows that described by Kulmala et al. (2006). Gaseous sulphur compounds (sulphur dioxide, $SO_2$ and dimethyl sulphide, DMS) and VOCs are oxidised, forming low volatility gases which condense irreversibly onto pre-existing aerosol. Condensation is calculated following Fuchs (1971) which is described in Mann et al. (2010). Mineral dust is treated by the model in a separate aerosol module (Woodward, 2001).

The emissions used are all decadal-average emissions, centred on the year 2000, and monthly varying.  Anthropogenic and biomass burning gas-phase emissions are prescribed following Lamarque et al. (2010). Biogenic emissions of isoprene, monoterpene and methanol ($CH_3OH$) are also prescribed, taken from the Global Emissions Inventory Activity (GEIA), based on Guenther et al. (1995). A diurnal cycle in isoprene emissions is imposed. POA and BC emissions from fossil fuel combustion are prescribed following Lamarque et al. (2010). POA and BC emissions from savannah burning and forest fires are prescribed, taken from the Global Fire Emissions Database (GFEDv2 (van der Werf et al., 2010)). All carbonaceous emissions are emitted into the insoluble mode and are transferred to the soluble mode by condensation ageing. Ageing proceeds at a rate consistent with a 10-monolayer coating being required to make a particle soluble.

## 2.1 Formation of SOA in the standard version of the model

In UKCA, VOCs undergo oxidation ([o] = OH, $O_3$ and $NO_3$). VOC oxidation products considered with a low enough volatility to condense are represented by a single surrogate compound, SOG. The reaction yield (α) describes the molar quantity (stoichiometric coefficient) of low volatility vapours formed.  SOG condenses irreversibly onto the surface of pre-existing aerosol, calculated following Fuchs (1971).

$$VOC + [o] \rightarrow \alpha\,SOG \rightarrow SOA \qquad\qquad\qquad (1)$$

In this study, as with the majority of global aerosol models (e.g. Tsigaridis et al., 2014), OA is treated as non-volatile, species, the emission or chemical yield implicitly reflecting only the particle phase component of OA. As we discuss in Section 1, anthropogenic OA in particular has a substantial semi-volatile component, which will introduce an important mode of variability to aerosol properties in industrial regions. In UKCA, monoterpenes ($C_{10}H_{16}$), are the only VOC considered in SOA formation. Monoterpene is oxidised, following reaction kinetics taken from Atkinson et al. (1989). The reaction yield applied to monoterpenes was assumed to be 13 %, which is identical to other global modelling studies, (Mann et al., 2010;Scott et al., 2014;Scott et al., 2015) , which was taken from (Tunved et al., 2006) who estimate the yield at 10 – 13 %. Global annual monoterpene emissions are 142 Tg (monoterpene) $a^{-1}$ and their spatial distribution is shown in Figure 1.

**2.2 New SOA sources**

For this study, new SOA sources are implemented in UKCA. Specifically, isoprene and lumped anthropogenic and biomass burning VOCs are added as precursors of SOA. Biogenic isoprene emissions are taken from the GEIA database (Guenther et al., 1995) and are equal to 561 Tg (isoprene) $a^{-1}$. Isoprene reaction kinetics are taken from Atkinson et al. (1989). For the biomass burning source of SOA, CO emissions from biomass burning were used to define its spatial distribution (Lamarque et al., 2010) and scaled to reproduce the global annual VOC total emissions from biomass burning estimated from Emissions Database for Atmospheric Research (EDGAR) (49 Tg (VOC) $a^{-1}$). The biomass burning source of SOA is hereafter referred to as $VOC_{BB}$. Anthropogenic emissions of aromatic compounds, benzene, dimethylbenzene and trimethylbenzene, were taken from Lamarque et al. (2010), and used together to define the spatial distribution for the anthropogenic source of SOA. These were scaled to reproduce the global annual anthropogenic VOC total emissions estimated by EDGAR (127 Tg (VOC) $a^{-1}$). This represents the anthropogenic source of SOA and will be referred to as $VOC_{ANT}$.

The anthropogenic and biomass burning VOCs are lumped species, which results in difficulty in selecting reaction kinetics for these species. The VOCs released from anthropogenic and biomass burning having a range of carbon-carbon bonding types, with a range of carbon chain length, and a range of functional groups. The exact speciation of these mixtures has not been resolved, especially for higher molecular weight species (Yokelson et al., 2013). However, more recent measurements of biomass burning (Stockwell et al., 2015;Hatch et al., 2017) and vehicle fuel (May et al., 2014;Zhao et al., 2015b) emissions in laboratory conditions reveal substantial quantities of oxygenated aliphatic, aromatic (e.g. benzene, toluene, etc.), polycyclic aromatic (e.g. naphthalene), furans, as well as large fractions of unknown species. Considering this range in chemical speciation, two compounds were used to represent the reactivity of the anthropogenic and biomass burning precursors in this study – naphthalene and monoterpene. For all simulations, $VOC_{ANT}$ and $VOC_{BB}$ are assumed to react solely with OH. Initially, $VOC_{ANT}$ and $VOC_{BB}$ are assumed to have identical reactivity to monoterpene. As monoterpene is a relatively reactive species, this provides an upper estimate for the rate for anthropogenic and biomass burning VOC oxidation. A lower estimate of SOA production from $VOC_{BB}$ is provided by assuming reactivity of naphthalene. Naphthalene has been used as surrogate compound

for IVOCs (Pye and Seinfeld, 2010) and is roughly 50 % less reactive than monoterpene. Both monoterpene and naphthalene species are used to represent the reactivity of $VOC_{ANT}/VOC_{BB}$ as they provide relatively wide estimates of the reactivity of these surrogate compounds. For all the new species added to SOA production: isoprene, $VOC_{ANT}$ and $VOC_{BB}$, initially, a reaction yield of 13 % is applied. As discussed in Section 1, reaction yields vary from one study to another, as well as within

individual studies. Furthermore, $VOC_{ANT}$ and $VOC_{BB}$ are surrogate compounds, representing a mixture of species. The initial assumption of identical reaction yields for all species does not negate the findings from laboratory studies, which suggest reaction yields are highly dependent on both molecular structure. However, the substantial uncertainties of reaction yields, coupled with these species representing lumped species, prevents accurate selection of laboratory-derived reaction yields for specific compounds. In addition, identical reaction yields allows differences in SOA concentrations to be solely attributed to

differences in the spatial pattern, seasonality, and magnitude of VOC precursor emissions. However, the influence of accounting for differences in reaction yields is explored in two additional simulations described below; the reaction yields for isoprene is assumed to be 3 %, which is suggested by (Kroll et al., 2005, 2006). Also, the reaction yield for $VOC_{ANT}$ iIn this studuys increased from 13 to 40 %, which is motivated by the widespread model negative bias in urban environments among global modelling studies.

    The spatial pattern of precursor emissions from these additional SOA sources is also shown in Figure 1. The seasonal cycle of the global precursor emissions from all of the VOC sources is shown in Figure 2. Biogenic and biomass burning emissions peaking during NH summer, whereas anthropogenic emissions are highest during NH spring and winter. Rate coefficients are taken from Atkinson et al. (1989) and are summarised in Table 1.

**2.3 - Model simulations**

Eight model simulations were performed using the different VOC sources of SOA for two years 1999-2000. The first year was discarded as spin up and the analysis is based on the year 2000 (Table 2). For all SOA components and across all simulations, SOA is solely removed by wet and dry deposition. The control simulation (Control) uses monoterpene as the only SOA precursor. Isoprene (its biogenic terrestrial source only), $VOC_{BB}$ and $VOC_{ANT}$ are introduced in additional simulations: Iso,

BB and Ant respectively. All sources of SOA are combined in the AllSources simulation. A number of sensitivity simulations were also carried out. The first sensitivity study tests a lower isoprene reaction yield of 3 % (Iso (3%)) which is suggested by laboratory data (Kroll et al., 2005, 2006). The second sensitivity study tests a higher $VOC_{ANT}$ reaction yield of 40 % (Ant (40%)) which is suggested by (Spracklen et al., 2011). Another study has also investigated such a large anthropogenic SOA source, however, this was done by scaling simulated anthropogenic SOA concentrations (Heald et al., 2011). A final sensitivity

study tests the influence of the assumed reactivity of VOCBB on SOA. Here, with a reaction yield of 13 %, the reactivity is assumed to be identical to naphthalene. Naphthalene was chosen as it has been used as a surrogate compound to represent

IVOCs in a global modelling study (Pye and Seinfeld, 2010) and it is roughly 50 % less reactive than monoterpene (Atkinson and Arey, 2003).3 Observations used to evaluate modelled OA

## 3 Observations

All measurements of OA mass concentrations used to evaluate the model were made using the Aerosol Mass Spectrometer
(AMS) (Canagaratna et al., 2007;Jayne et al., 2000). Uncertainties in the observed OA mass concentration are estimated to be ~30–35% (Bahreini et al., 2009). Observations can be accessed on the AMS global network website 2.2)(https://sites.google.com/site/amsglobaldatabase/)

Surface OA observations from the AMS network, originally compiled by Zhang et al. (2007), span the time period 2000-2010. The 37 observed surface measurement locations are shown in Figure 3 and coloured according to the environment
sampled from: urban, urban downwind, or remote. Note, for some sites, multiple measurements were made at different time periods. Surface OA mass concentrations were analysed further using positive matrix factorisation (PMF) to classify OA as either oxygenated OA (OOA) or hydrocarbon-like OA (HOA). We assume observed OOA is comparable to simulated SOA, and observed HOA is comparable to simulated POA. This assumption is made in several other studies (Hodzic et al., 2016;Tsimpidi et al., 2016). The dataset is supplemented with 3 additional observations of total OA over Santiago (Chile)
(Carbone et al., 2013), Manaus (Brazil) (Martin et al., 2010), and Welgegund (South Africa) (Tiitta et al., 2014) These were included as they expand the geographical coverage over which the model can be evaluated. To compare UKCA results to observations spatially, the nearest model grid box (based on its centre co-ordinates) to the measurement site location was selected. The month in which the observations fall is compared to the simulated monthly mean concentration for the year 2000. Note that comparing simulated OA in the year 2000 with measured OA from a different year may introduce discrepancies for
model-observation comparison. This is particularly important for regions influenced by biomass burning as interannual variability of biomass burning emissions is extremely high (Tsimpidi et al., 2016). However, simulating the entire measurement period was not possible.

Aircraft OA concentrations from the AMS network, originally compiled by Heald et al. (2011), are also shown in Figure 3. For details and references of aircraft campaign, see https://sites.google.com/site/amsglobaldatabase/. These measurements
originate from ten intensive campaigns worldwide over the period 2000 - 2010. Four campaigns were carried out in remote regions, located over the north Atlantic Ocean (TROMPEX and ITOP), Borneo (OP3) and the tropical Pacific Ocean (VOCALS-UK). Three campaigns were carried out in North America and were influenced heavily by biomass burning (ARCTAS-A, ARCTAS-B and ARCTAS-CARB). Three campaigns were also carried out in polluted regions of Europe (EUCAARI, ADIENT and ADRIEX). To compare against UKCA, aircraft observations were binned onto the model's vertical
grid. The nearest model grid box to the average horizontal aircraft measurement location was selected for comparison. Again the month of the observations were matched to the monthly mean estimate for 2000 simulated by UKCA. As with the surface evaluation, note the mismatch in measurement and simulation years as a potential contributor to model-observation discrepancies.

## 4 Results and discussion

### 4.1 Global SOA budget

In this section, the effects of individual sources of SOA on the global SOA budget are evaluated. Table 3 shows simulated annual SOA production rates for the simulations described in Table 2, as well as estimates taken from the literature. When monoterpene is the only source of SOA, the annual global SOA production rate is 19.9 Tg (SOA) a$^{-1}$. With reaction yields of 13 %, the inclusion of isoprene (Iso), VOC$_{BB}$ (BB) and VOC$_{ANT}$ (Ant) increases the annual global SOA production rate by 19.6, 9.5 and 24.6 Tg (SOA) a$^{-1}$, respectively. Differences in SOA production rates for the different sources of SOA are due to differences in the spatial and temporal variability of VOCs and oxidant concentrations, as well as differences in reaction constants. For example, surface oxidant concentrations are extremely spatially variable, with relatively lower concentrations over the Amazon and Congo, and higher concentrations over continental regions of the NH. Additionally, the short-lived oxidants, OH and NO$_3$, have strong diurnal profiles. In contrast, O$_3$ is more temporally homogeneous. VOC emissions are also spatially variable; in tropical forests, emissions of isoprene exceed emissions of monoterpene, whereas, in boreal forests, emissions of monoterpene exceed emissions of isoprene. The response of bVOCs to regional and temporal oxidant concentration variability, together with differences in reaction constants for oxidation (Table 1) explains why SOA production from isoprene, despite its higher emissions compared to monoterpene has similar SOA production rates.

With isoprene and monoterpene as sources of SOA, the global annual biogenic SOA production rate is 39.5 Tg (SOA) a$^{-1}$ (Table 3). This is in reasonable agreement with estimates of biogenic SOA production from most other global modelling studies (14.9 – 55 Tg (SOA) a$^{-1}$; Table 3) despite possible differences in which biogenic VOCs are included in the SOA schemes. One global modelling study suggests an annual global biogenic SOA production rate of 97.5 Tg (SOA) a$^{-1}$ ((Hodzic et al., 2016); Table 3) based on reaction yields that account for wall losses during chamber studies. In contrast, an observationally-constrained top-down estimate of biogenic SOA production (13 Tg (SOA) a$^{-1}$; Table 3) is much lower than our estimate. In our sensitivity simulation, when the reaction yield describing SOA formation from isoprene is reduced to 3 %, the annual global biogenic SOA production rate decreases to 23.9 Tg (SOA) a$^{-1}$ (Table 3), which is still consistent with other global modelling studies.

With a reaction yield of 13 %, the annual global SOA production rate from anthropogenic sources is 24.6 Tg (SOA) a$^{-1}$ (Table 3). This is higher than other global modelling studies (range of 1.6 – 19.2 Tg (SOA) a$^{-1}$) but substantially lower than the observationally-constrained top-down estimate (100 Tg (SOA) a$^{-1}$) from Spracklen et al. (2011). In our sensitivity simulation when SOA formation from anthropogenic sources is increased from 13 to 40 %, the annual global SOA production rate increased to 70.0 Tg (SOA) a$^{-1}$. Compared to other sources, biomass burning is the smallest source of SOA, yet still significant (9.5 Tg (SOA) a$^{-1}$; Table 3). The magnitude of SOA production from biomass burning is consistent with observations (1 - 15 Tg (SOA) a$^{-1}$; Table 3) and top-down studies (3 – 34 Tg (SOA) a$^{-1}$; Table 3). However, biomass burning SOA production rates vary substantially from different modelling studies. The reason for such large differences between Hodzic et al. (2016) and Shrivastava et al. (2105), both of which simulate biomass burning SOA formation from S/IVOCs,

could be due to the lack of knowledge of S/IVOCs emissions and chemistry, resulting in the need for assumptions when including SOA formation from these species in models (Shrivastava et al., 2017). When all sources of SOA are included with identical reaction yields, the annual global SOA production rate is 74.0 Tg (SOA) $a^{-1}$. This lies within the range of other estimates based on bottom-up methods (13 – 132 Tg (SOA) $a^{-1}$; Table 3). However, top-down approaches, such as scaling of

the sulphate budget (Goldstein and Galbally, 2007), or constraining the SOA budget by satellite (Heald et al., 2010) or in-situ observations (Spracklen et al., 2011), are substantially greater (50 – 1820 Tg (SOA) $a^{-1}$; Table 3).

   With identical reaction yields, the annual-average global SOA burden, from monoterpene, isoprene, biomass burning and anthropogenic precursors is 0.19, 0.22, 0.13 and 0.38 Tg (SOA), respectively. With monoterpene only, the annual average global lifetime of SOA is 3.5 days. Inclusion of isoprene and biomass burning as sources of SOA has little effect on the SOA

lifetime, with annual-average global lifetime of SOA ranging from 3.7 – 4.0 days in these simulations. However, inclusion of an anthropogenic source of SOA increases the SOA lifetime to 4.7 days, and to 5.1 days with an anthropogenic source with a 40% yield. The effect of new sources of SOA on SOA lifetime suggest that SOA from monoterpene, isoprene and biomass burning has a similarly short lifetime, whereas SOA from anthropogenic sources has a relatively longer lifetime.

   The variation in lifetime for the different SOA components is likely due to differences in the spatial distributions of SOA

precursor emissions, as well as the extent of co-location of emissions and precipitation. Biogenic and biomass burning VOCs, primarily located in tropical forest regions of the southern hemisphere, experience different precipitation rates compared to anthropogenic VOCs, which are primarily released in urban and industrial regions of the northern hemisphere. Vertical gradients in SOA production can also affect the SOA lifetime. However, in this study, all SOA precursors are emitted at the surface hence, the various SOA components in this study likely have very similar vertical gradients. Shrivastava et al. (2105)

find that the SOA lifetime substantially increases when biomass burning precursors are emitted at higher altitudes. The range in SOA lifetimes over the different simulations in this study is in agreement with Tsigaridis et al. (2014), which ranged from 2.4 – 15 days. These SOA lifetimes are also in good agreement with Hodzic et al. (2016) who estimate the SOA lifetime from biogenic VOCs, anthropogenic and biomass burning VOCs combined, and anthropogenic and biomass burning S/IVOCs to be 2.2, 3.3 and 3.0 days, respectively.

The simulated global annual cycle of SOA production and SOA burden from varying sources of SOA is shown in figure 4. Figure 4 shows biogenic and biomass burning SOA production peaking during NH summer (Figure 4 a). This is due to both high emissions (Figure 2) and elevated photochemistry during this season. This results in both biogenic and biomass burning SOA burdens also peaking during this season (Figure 4 b). The seasonal cycle of the biomass burning SOA is consistent with Tsimpidi et al. (2016). The global SOA production rate and the SOA burden were also found to peak during NH summer in

the multi-model study by Tsigaridis et al. (2014); however the seasonal cycle of SOA production and SOA burden for different SOA components could not be determined. The anthropogenic emissions peak during NH spring and winter (Figure 2), and are offset by the seasonal cycle of photochemistry (that influences oxidation), resulting in constant anthropogenic SOA production all year round (Figure 4 a). However, the anthropogenic SOA burden shows a pronounced seasonal cycle with a double peak during spring and autumn and a minimum during summer (Figure 4 b). Therefore, the seasonality of the

anthropogenic SOA burden must be driven by the seasonality of the anthropogenic SOA lifetime, since it differs from the seasonal cycle of SOA production. Indeed, anthropogenic SOA precursor emissions are highest over China and India (Figure 1). Over these regions, during summer, rainfall is greatest, which may result in a reduction in SOA lifetime due to greater wet deposition and a decrease in SOA burden. The seasonal profile of the global anthropogenic SOA burden is in agreement with that found by Tsimpidi et al. (2016); however, they suggest an alternative cause. In their model, SOA is treated as semi volatile and can partition between the aerosol and gas-phase (as opposed to in this study, where SOA is treated as non-reactive and non-volatile). In their study, partitioning is dependent on a number of parameters, including temperature. Tsimpidi et al. (2016) suggest that the summertime peak in photochemistry is compensated for by enhanced SOA evaporation. Further research is required to quantify the relative importance of each mechanism on the SOA burden.

## 4.2 Effects of new SOA sources on simulated SOA and POA spatial distributions

In this section, the effects of new sources of SOA on both surface SOA and POA distributions are explored. Figure 5 shows the annual-average surface SOA concentrations simulated with a monoterpene source of SOA alone (Control) and with all sources of SOA (monoterpene, isoprene, $VOC_{BB}$ and $VOC_{ANT}$) (AllSources) (Table 1). In the monoterpene only simulation, annual-average surface SOA concentrations range between 3 - 6 µg m$^{-3}$ over tropical forest regions of South America and Africa, as well as lower SOA concentrations of up to 3 µg m$^{-3}$ in the South East USA, and in parts of northern India, China and South East Asia (Figure 5 a). These patterns generally reflects the location of peak monoterpene emissions (Figure 1), which are emitted by vegetation only in the emissions inventory used here. Fast SOA production from monoterpene and a relatively short lifetime of SOA results in SOA concentrations peaking at the emissions source with a sharp decrease downwind. The addition of new SOA sources (isoprene, $VOC_{BB}$ and $VOC_{ANT}$) together alongside monoterpene roughly doubles annual-average surface SOA concentrations compared to simulations with monoterpene only. In particular, over the Amazon and the Congo region, annual average surface SOA concentrations peak between 5 and 10 µg m$^{-3}$ (Figure 5 b). Over industrialised and urban regions of India and China, annual-average surface SOA concentrations exceed 10 µg m$^{-3}$. Over large parts of the USA, Europe and most of Asia, SOA concentrations exceed 0.6 µg m$^{-3}$ (Figure 5 b).

The spatial patterns of SOA concentrations associated with each individual SOA source and all three new sources combined are highlighted in Figure 6, which shows the difference in annual-average surface SOA concentrations for each separate SOA source relative to the monoterpene only case. The inclusion of all new sources of SOA increases annual-average surface SOA concentrations substantially (1-10 µg m$^{-3}$) in both the NH and southern hemisphere (SH), and over continental regions as well as to a lesser extent over downwind oceanic regions (up to 1 µg m$^{-3}$) (Figure 6a). Isoprene is the most abundant VOC in the atmosphere globally, therefore, including this species in SOA production results in a substantial increase (3-10 µg m$^{-3}$) in SOA concentrations, especially in tropical regions (Figure 6b). Both monoterpene and isoprene contribute comparably to SOA concentrations (Figure 6b, Figure 5a). Over the Amazon and Congo, the two biogenic sources of SOA (monoterpene and isoprene) produce SOA concentrations that typically range from 3 – 10 µg m$^{-3}$ (Figures 6b, 5a), which is in agreement with other studies (Hodzic et al., 2016;Shrivastava et al., 2015;Farina et al., 2010). However, (Spracklen et al., 2011) suggests

that biogenic sources yield only a small amount of SOA (global production = 13 Tg a$^{-1}$; Table 3) and, therefore, simulated SOA concentrations in this region in their study did not exceed 1 µg m$^{-3}$ (Spracklen et al., 2011).

Biomass burning SOA also peaks in tropical forest regions of South America and the Congo region of Africa (Figure 6 c), corresponding to regions of intense forest and savannah fires (Figure 1). Over this region, annual average surface SOA
concentrations from biomass burning typically range from 1-3 µg m$^{-3}$ (Figure 6 c) and have lower values compared to SOA concentrations arising from the two biogenic sources. Biomass burning also contributes 0.2 – 1 µg m$^{-3}$ to annual-average surface SOA concentrations over boreal forests of northern China and Eastern Siberia. The location and magnitude of these peak SOA concentrations is in agreement with other global modelling and observationally-constrained studies (Tsimpidi et al., 2016;Spracklen et al., 2011) despite differences in biomass burning production rates (Table 3), highlighting the importance of
SOA lifetimes in determining SOA concentrations.

The inclusion of an anthropogenic source of SOA increases SOA concentrations over much of the NH. Over industrialised and urban regions of China and India, annual-average surface anthropogenic SOA concentrations typically exceed 6 µg m$^{-3}$ (Figure 6 d). The location of peak annual-average surface anthropogenic SOA concentrations, which is reflected by the location of anthropogenic combustion emissions (Figure 1), is in agreement with other global modelling studies (Spracklen et al.,
2011;Tsimpidi et al., 2016;Hodzic et al., 2016). The magnitude of peak SOA concentrations are lower than Tsimpidi et al. (2016) but consistent with that of Spracklen et al. (2011). The inclusion of an anthropogenic source of SOA also results in increases in annual-average surface SOA concentrations in remote regions: between 0.1-0.6 µg m$^{-3}$ over the North Atlantic and Pacific Oceans with larger values across the Indian Ocean (>1µg m$^{-3}$).

The difference in annual-average surface SOA concentrations for the isoprene and anthropogenic sensitivity simulations,
relative to the monoterpene only simulation are shown in Figure 7. In the isoprene sensitivity simulation, reducing the reaction yield decreases the proportion of oxidation products which are condensable, therefore lowering SOA concentrations (c.f. Figure 7 a; Figure 6 b). Hence, SOA concentrations are only slightly elevated in this simulation compared to the monoterpene only simulation, leading to biogenic SOA concentrations over Amazon and Congo of ~10 µg m$^{-3}$. In addition, for a decrease in reaction yield from 13 to 3 % (factor of 4.3) the global annual-average surface SOA concentration from isoprene reduces
by a factor of 4.25. This suggests that SOA concentrations, at least for the global mean, respond linearly to changes in reaction yields. In the anthropogenic sensitivity simulation, the reaction yield is increased such that the annual average surface anthropogenic SOA concentrations increases by up to 17 µg m$^{-3}$ across most industrialised regions relative to the monoterpene only simulation (Figure 7 b). The magnitude of these peak SOA concentrations are in broad agreement with Tsimpidi et al. (2016). Contrastingly, our peak simulated anthropogenic SOA concentrations exceed those from the Spracken et al. (2011)
study, despite our smaller production rate (Table 3), again, suggesting the importance of differences in SOA lifetime in determining SOA concentrations. For an increase in reaction yield of a factor of three, surface SOA concentrations increase by the same amount, further corroborating the linear dependence of surface concentrations on reaction yield, as observed in the isoprene sensitivity simulation.

The SOA spatial distributions simulated in this study may be sensitive to the assumption of fast reaction kinetics for anthropogenic and biomass burning SOA precursors. Here, we have assumed anthropogenic and biomass burning sources of SOA are oxidised on a timescale identical to that of monoterpene. This is due to limited information on the identity of dominant SOA precursors from these sources. The influence of assumed reactivity on simulated SOA from biomass burning was investigated in an additional sensitivity simulation where $VOC_{BB}$ adopts the reactivity of naphthalene (Table 2; section 2.3) , an aromatic species which has been used as a surrogate compound for IVOCs (Pye and Seinfeld, 2010). Compared to monoterpene, naphthalene is roughly 50 % less reactive (Atkinson and Arey, 2003). However, despite this substantial reduction in reactivity, the global annual-total SOA production rate from biomass burning VOCs is reduced by less than 1 %. Also, the simulated spatial distributions are almost identical for the two $VOC_{BB}$ species. Like all other SOA precursors in this study, $VOC_{BB}$ does not undergo dry or wet deposition. Therefore, a reduction in the reactivity of $VOC_{BB}$ does not affect the fate of this compound.

Next, the effects of new sources of SOA on simulated POA concentrations are explored. Figure 8 shows the annual-average surface POA concentrations simulated with monoterpene as the only VOC source. The emissions inventory used here includes POA emissions from both biomass burning and anthropogenic sources. Within UKCA, all emitted POA is assumed to be hydrophobic. Soluble vapours, such as sulphate and organic compounds (represented as SOG), condensing onto the surface of POA particles, transfer hydrophobic POA particles into the hydrophilic mode (condensation ageing). UKCA assumes 10 monolayers of soluble material are required to redistribute hydrophobic particles into the hydrophilic mode. Over the Amazon, total POA lies in the range of $0.34 - 2.2$ µgm$^{-3}$ and is almost entirely hydrophilic (Figure 8a and b). This is due to sufficiently high SOA concentrations in the monoterpene-only source simulation, which also peak in this same region (Figure 5 a). Over the Congo region, total POA concentrations lie between $1.2 - 4$ µgm$^{-3}$ (Figure 8a) and hydrophobic POA concentrations range from 0.17 to 1.51 µgm$^{-3}$ (Figure 8b). In this region, SOA concentrations are high enough (Figure 5 a) to re-distribute the majority of POA into the hydrophilic mode, however, a small amount remains in the hydrophobic mode. Over northern China and eastern Siberia, total POA concentrations are extremely high, ranging from 4 to 25 µgm$^{-3}$ (Figure 8 a). In this region, SOA concentrations are low (Figure 5 a), therefore, a substantial fraction of POA remains in the hydrophobic mode (range $2.59 - 13.2$ µgm$^{-3}$ (Figure 8 b). Overall, 25 % of the global annual-average POA burden is hydrophobic.

When new sources of SOA are added to the model, condensation-ageing of POA increases and the proportion of hydrophilic POA increases. Critically, wet deposition removes hydrophilic particles but not hydrophobic particles. Therefore, inclusion of new sources of SOA decreases the POA lifetime and results in decreased POA concentrations. Figure 9 shows the difference in annual-average surface POA concentrations relative to the control simulation. Over the Congo region, inclusion of isoprene and biomass burning as sources of SOA results in a decrease in annual average surface POA concentrations of greater than 3 µgm$^{-3}$ (Figure 9a and b). In this region, the new sources of SOA enhances POA transfer from the hydrophobic to hydrophilic modes, which are efficiently removed by deep tropical convection. Over Eastern Siberia, the inclusion of a biomass burning source of SOA also decreases POA concentrations.

Over the Amazon, although inclusion of isoprene and biomass burning results in substantial increases in SOA concentrations (Figure 6 b and c), there are negligible changes in POA concentrations (Figure 9). In this region of the world, in the monoterpene only simulation, the majority of POA is hydrophilic. For this reason, increased SOA concentrations in this region have no effect on the partitioning of POA between the hydrophobic and hydrophilic modes, hence, there is little change in POA lifetime. Over urban and industrialised regions of India and China, hydrophobic POA concentrations are much higher (Figure 8b). In these regions, the inclusion of isoprene and anthropogenic sources of SOA results in substantial increases in SOA concentrations (Figure 6 b and d), therefore, hydrophilic POA concentrations in this region are increased. However, there are minimal changes in annual-average surface total POA concentrations (Figure 9) which is probably due to inefficient wet removal. Across all simulations, in various locations, inclusion of new sources of SOA reduces POA concentrations. However, the decrease in POA concentrations is always outweighed by the increase in SOA concentrations, thus, total OA increases.

To summarise, with monoterpene emissions only, SOA concentrations peak in the SH over tropical forest regions of South America and Africa. Including isoprene, biomass burning and anthropogenic sources of SOA results in substantial increases in SOA concentrations. Isoprene and biomass burning sources of SOA produce substantial increases in SOA concentrations over the Amazon and Congo compared to the monoterpene only source. The anthropogenic source of SOA increases SOA concentrations over industrialised and urban regions of China, India, USA and Europe. Sensitivity studies show that SOA concentrations respond linearly to changes in reaction yields. Upon inclusion of new SOA sources, increased SOA concentrations lead to decreased POA concentrations, however, in all regions of the globe, modelled total OA increases.

**4.3 Effects of new SOA sources on model agreement with observations**

In this section, simulated SOA, POA and total OA concentrations (see Section 4.2), are evaluated against observations (see Section 3). First, simulated SOA and POA concentrations are compared to surface measurements. Next, to expand the spatial coverage of evaluation, simulated total OA concentrations are compared against surface observations. Finally, vertical profiles of simulated total OA concentration are compared against aircraft campaign observations.

**4.3.1 Evaluation of surface SOA and POA concentrations**

Surface observations used to evaluate the model are shown in Figure 3. Note that the overall number of sites is small and the measurement locations are primarily in NH mid-latitude regions where anthropogenic emissions are high (Figure 1). However, these sites sample urban, urban downwind, and remote environments over Europe, North America and Asia. The mean and normalised mean bias (NMB) are used to evaluate model agreement with observations. A summary of statistics evaluating simulated SOA, POA and OA are shown in Tables 4, 5 and 6, respectively.

High SOA concentrations observed in urban environments (mean = 4.76 $\mu$g m$^{-3}$) are maintained further downwind (mean = 3.93 $\mu$g m$^{-3}$) and in remote environments (mean = 2.70 $\mu$g m$^{-3}$) (Table 4). Contrastingly, observed POA concentrations peak in urban environments (mean = 2.79 $\mu$g m$^{-3}$) but decrease rapidly further downwind (mean = 0.78 $\mu$g m$^{-3}$) to almost negligible values in remote environments (mean = 0.14 $\mu$g m$^{-3}$). Zhang et al. (2007) suggest that cities act as sources of POA, whereas,

both cities and remote environments are sources of SOA. Compared to other cities, observed SOA and POA are extremely high in densely populated cities. For example, observed SOA concentrations in Beijing and Mexico City are 17.3 and 14.55 µg m$^{-3}$, respectively, roughly 3 times greater than the mean observed SOA in urban environments. Observed POA concentrations in Beijing and Mexico City are 7.4 and 7.23 µg m$^{-3}$, respectively, roughly 2.5 times greater than the mean observed POA in urban environments.

Figure 10 compares simulated SOA and POA from the monoterpene only simulation against AMS measurements. When considering all observations, simulated SOA is substantially lower than observed (NMB = -91 %) (Table 4, Figure 10 a) whereas simulated POA is in better but still relatively poor agreement with observed POA (mean = 0.71 µg m$^{-3}$, NMB = -43 %) (Table 5, Figure 10 b). A model negative bias in SOA and POA is common among global models (Tsigaridis et al., 2014). It is suggested that this underestimate is partly due to the coarse grid resolution, which is unable to resolve both urban-scale pollution and heterogeneities in remote environments (Kaser et al., 2015), but for SOA, uncertainties in its sources and their reaction rates, as highlighted in section 3.1, will also be important. The simulated negative bias in SOA occurs for all site type environments (NMB range -85 to -93 %) and all continental regions (NMB range -83 to -98 %) (Table 4). In the case of POA, the model has a negative bias that is larger in urban compared to urban downwind environments. This may indicate that the POA emissions inventory used by the UKCA model underestimates anthropogenic POA emissions. Known, missing sources of POA from the emissions inventory used here include cooking OA and S/IVOCs. Emissions of OA from residential and commercial cooking activities have been measured to contribute 17 – 19 % to total OA in urban environments (Hayes et al., 2013;Mohr et al., 2012;Ge et al., 2012). S/IVOCs can also contribute to POA (Robinson et al., 2007). The amount of S/IVOCs missing from traditional emissions inventories is estimated to be 0.25 – 2.8 times traditional POA emissions (Shrivastava et al., 2008;Robinson et al., 2010). Regionally, over Asia, simulated POA is in good agreement with observations (NMB= 2%; Table 5). However, the model underestimates POA over Europe (NMB = -59 %) and North America (NMB = -70 %) (Table 5). Underestimated POA concentrations over Europe have been reported in previous global modelling studies, and attributed to under estimated emissions from residential biofuel and biomass burning in residential areas (van der Gon et al., 2015).

In remote environments, simulated POA is overestimated compared to observations (mean = 0.70 µg m$^{-3}$, NMB = 410 %) (Table 5, Figure 10 b). This may be due to the assumption that POA is non-volatile and unreactive in UKCA (i.e. missing sinks). Similarly, Spracklen et al. (2011) also treats POA as non-volatile and unreactive and, consequently, overestimates POA compared to observations in remote environments. By considering heterogeneous POA oxidation to form SOA, the NMB against observed POA reduces from 274 to 45 % (Spracklen et al., 2011). Additionally, Tsimpidi et al. (2016) allows POA to oxidise to form SOA via the gas phase and finds relatively good agreement between simulated and observed POA concentrations in remote environments.

Finally, the impact of new sources of SOA on model agreement with observations is explored. Figure 11 shows simulated SOA against observations for the simulations that include all the SOA sources and the individual sources in addition to monoterpene (Iso, Ant, BB; Table 2). When considering all the measurement site data, the inclusion of all new sources of SOA reduces the model negative bias in surface SOA concentrations compared to observations from -91 to -50%  (Figure 11 a,

Table 4). This improvement in the model negative bias is primarily due to the inclusion of the anthropogenic source of SOA (NMB = -65 %) (Figure 9 d, Table 4). This is because the anthropogenic source of SOA generates high SOA concentrations (Figure 6 d) in areas with a high density of observations (Figure 3). When restricting observations classified by environments (urban, urban downwind and remote) or continent (i.e. Europe, North America and Asia), the reduction in the model negative

bias when all sources of SOA are included is also mainly due to the anthropogenic source of SOA. The inclusion of isoprene and biomass burning as sources of SOA, reduces the NMB by only 7 and 1 %, respectively, compared to the monoterpene source only results (Figure 11 b and c, Table 4). Although simulated SOA concentrations from both isoprene and biomass burning are high (Figure 6 b and d), generally, peak concentrations associated with these sources do not occur in locations with measurements of SOA (Figure 3).

Figure 12 shows simulated versus observed SOA concentrations for the isoprene and anthropogenic sensitivity simulations with reaction yields of 3 % and 40 %, respectively. When the reaction yield of anthropogenic SOA formation is increased from 13 to 40 %, model negative biases are reduced further. When all sites are considered, simulated SOA concentrations are in fairly good agreement with observations (NMB = -10 %), highlighting a large influence from anthropogenic sources of SOA at these measurement site locations, and the requirement of a higher reaction yield than 13% to match observed levels of SOA.

However, simulated SOA concentrations are still underestimated in urban environments (NMB = -34 %) and slightly overestimated further downwind (NMB = 14 %) and at remote site locations (NMB = 12 %). This positive bias downwind and in remote environments could be due to an overestimated SOA lifetime. Hodzic et al. (2016) suggests that global models typically overpredict SOA lifetime by low SOA wet deposition rates. When SOA scavenging efficiency was increased in their model simulations, the SOA lifetime decreased from 6 − 10 days to 2.2 − 3.3 days, and positive biases in simulated SOA

concentrations downwind were reduced (Hodzic et al. (2016)). With a reaction yield of 40% for anthropogenic SOA formation, simulated SOA is slightly overestimated over Europe (NMB = 12 %) and underestimated over North America and Asia (NMB= -12 to -17 %). For the isoprene sensitivity simulation, there is almost no change in the model negative bias at these NH mid-latitude measurement locations in comparison to the monoterpene only source simulation. As highlighted above, there are no measurements of SOA concentrations in tropical isoprene-sensitive regions for suitable model evaluation.

When considering POA concentrations, the agreement between simulated and observed POA is largely unchanged by the inclusion of new sources of SOA. With the new SOA sources, POA condensation-ageing increases, resulting in the newly soluble POA particles undergoing wet removal. This decreases POA lifetime and causes POA concentrations to decrease, as described above in relation to Figure 9. When considering all observations, the inclusion of new sources of SOA has a reduction in the model negative biaswith the NMB changing by ~2 % (Table 5); this is also the case for individual site

types and across the three continental regions.

        The observations used to evaluate SOA and POA concentrations thus far have been primarily located in the NH mid-latitudes. The geographical coverage over which the model is evaluated is expanded by including observations of total OA over the urban environment of Santiago (Chile) and the rural environments of Manaus (Brazil) and Welgegund (South Africa) (Figure 3). Figure 13 shows simulated OA for the different SOA sources against observed OA for the three additional non-

speciated OA measurements. Observed OA concentrations over Manaus (Brazil) and Welgegund (South Africa) are 0.77 and 3.49 µg m$^{-3}$, respectively. These concentrations are typical of remote environments in the NH mid-latitudes (mean = 2.83 µg m$^{-3}$; Table 6). Over Manaus (Brazil), out of all the new sources of SOA added to the model, the addition of isoprene as an SOA source has the largest impact on increasing simulated OA concentrations at this location. However, in the case of Manaus

(Brazil), OA concentrations are overestimated with the inclusion of isoprene as an SOA source (Figure 13b). This suggests that a lower isoprene reaction yield (Iso (3%)) may be more accurate (OA concentrations ~1.5 µg m$^{-3}$). Conversely, over Welgegund (South Africa) with only a monoterpene source of SOA, simulated OA is substantially lower than observed OA concentrations. Therefore, inclusion of new sources of SOA reduces the model negative bias at this location. Additionally, the anthropogenic sensitivity simulation has a substantial improvement in the model negative bias over Welgegund (South Africa)

suggesting that this downwind location is heavily influenced by anthropogenic emissions from the urban centre (Cape Town). This also appears to be the case for the urban downwind and remote sites in the NH mid-latitudes where the simulated OA concentrations for all sources and for anthropogenic sensitivity simulations yield the lowest model bias compared to observations (mean values for remote sites: 2.15 and 3.68 µg m$^{-3}$ respectively; Table 6).  Alternatively, Shrivastava et al. (2105) find that simulated OA at this site is primarily attributed to biomass burning.

Typical of densely populated cities, observed OA concentrations over Santiago are substantially higher than the modelled OA concentrations in all simulations. (Figure 13 a); which primarily reflects the difficulty of a coarse resolution global model to represent urban centres. Incorporation of these three observations expands the geographical coverage over which the model can be evaluated, especially over regions influenced by biogenic emissions. However, since these observations are not speciated, model biases in simulated OA concentrations cannot be attributed to SOA or POA

concentrations. Biases in simulated SOA and POA concentrations can sometimes act in unison (i.e. urban environments) or in competition (i.e. remote) with one another (Tables 4 and 5), hence the underlying processes are harder to discern. Additionally, there is a very low density of observations in these key regions of the world, therefore, the representativity of these sites for the region as a whole is unknown.

     In summary, when considering monoterpene as the only source of SOA, simulated SOA concnetrations are lower than

observed in all site types environments, as well as over North North America, Europe and Asia. Model performance is improved substantially when all new sources of SOA are included in the model, particularly due to the inclusion of an anthropogenic source of SOA. When the yield of anthropogenic SOA formation is increased, model agreement with observed SOA is further improved at all site types. However, there is now a very slight positive bias compared to observed SOA further downwind and in remote environments. When the spatial coverage of observations is expanded, to include measurements over

South America and Africa, isoprene becomes an important source of SOA. However, as very few observations have been made in this region. the observations cannot be used to robustly evaluate the effects of all the different  SOA sources. This highlights the need for more observations, particularly over regions of the world influenced by biogenic and biomass burning emissions.

### 4.3.2 Evaluation of OA vertical profile

In this section, simulated OA vertical profiles are compared to aircraft measurements from different campaigns that are described in Section 3 (Figure 3). The measurement campaigns cover different chemical environments in the atmosphere, sampling remote regions, regions influenced by biomass burning in North America and polluted regions in Europe.

Figure 14 shows the simulated OA vertical profile against the AMS aircraft measurements. Generally, observed OA concentrations peak in the lowermost kilometre and decline with altitude (e.g. EUCAARI, AIDENT, OP3 and TROMPEX). Biomass burning (ARCTAS campaigns; Figure 14 – bottom left) perturbs the vertical profile and results in elevated OA concentrations up to ~ 8 km. Observed OA concentrations in the ARCTAS campaigns are extremely high and extremely variable. Low OA concentrations from these campaigns reflect the remote regions of North America which are being sampled from. High OA concentrations from this campaign also reflect the plume-chasing approach, sampling directly from biomass burning plumes and resulting in extremely high OA concentrations.

The monoterpene only simulation typically underestimates observed OA concentrations in all environments and at all altitudes (Figure 14). This is common among global models when evaluated against aircraft campaigns, in which simulated SOA is purely biogenic (Utembe et al., 2011;Khan et al., 2017). The simulated OA from the monoterpene only simulation compares relatively well in remote environments (Figure 14- right hand side), typically lying within one standard deviation of the observations. When all SOA sources are included, OA concentrations increase, resulting in a smaller negative bias between simulated and observed OA concentrations at all altitudes. This is primarily due to the inclusion of the anthropogenic SOA source. However, simulated OA concentrations are larger than measured OA concentrations during the VOCALS and TROMPEX campaigns, especially at higher altitudes. The anthropogenic sensitivity simulation generally improves model agreement with aircraft observations even further for the polluted and biomass burning influenced campaigns, such that the simulated OA concentrations now generally fall within one standard deviation of the measurements at most altitudes (Figure 14 left hand side). In contrast, OA concentrations simulated for the remote campaigns, OP3 and TROMPEX, are now substantially overestimated compared to measurements away from the surface. This is in agreement with Heald et al. (2011) who found that a large anthropogenic source for SOA results in positive biases in simulated OA concentrations in remote environments. Other reasons for disagreement between model results and observations relate to our comparison methodology, whereby observed OA concentrations span the time period 2000-2010 and simulated OA concentrations are from the year 2000 (section 3). The inclusion of a biomass burning source of SOA has very little effect on model agreement with aircraft campaigns, even for campaigns influenced by biomass burning activity (ARCTAS). However, simulated biomass burning emissions peak over South America and Central Africa (Figure 1 c), whereas the aircraft campaigns influenced by biomass burning emissions were conducted in North America (Figure 3). Furthermore, biomass burning emissions vary substantially from year to year. Therefore, the mismatch in time period and the use of decadal mean emissions is particularly relevant for regions influenced by biomass burning. Higher temporal resolution emissions in biomass burning influenced regions may help model agreement. Indeed, the use of daily-varying fire emissions inventories has been shown to help reproduce observed OA

concentrations (Wang et al., 2011). In contrast, Shrivastava et al. (2105) find good agreement between simulated and observed OA when considering ARCTAS measurments, primarily due to their simulation of biomass burning SOA. These results highlight that biomass burning remains a highly uncertain source of SOA. Here, biomass burning SOA is considered from VOCs, with a global annual-total emission rate of 49 Tg (VOA) -1, and injected at the surface. Contrastingly, Shrivastava et al. (2105) treated biomass burning SOA from S/IVOCs, with global annual-total emissions of 450 Tg (VOC) $a^{-1}$, which are injected at the surface as well as at higher altitudes. Clearly, further research is required to identity the dominant sources of biomass burning SOA, as well emissions estimaates and chemistry.

## 5 Conclusions

Studies on different SOA sources are usually made in isolation and compared against different sets of observations, resulting in difficulty in drawing robust conclusions on the role of each SOA source on the SOA global budget and on model agreement with observations. In this study, a global chemistry and aerosol model (UKCA) was used to simulate SOA using all the known major sources of SOA comparing results to a consistent set of observations and examining their seasonal influence.

When monoterpene is the only source of SOA, the simulated annual global production rate is 19.9 Tg (SOA) $a^{-1}$. The inclusion of isoprene, biomass burning and anthropogenic sources of SOA increases the annual global SOA production rate by 19.6, 9.5 and 24.6 Tg (SOA) $a^{-1}$ respectively. When all sources are included, the simulated annual global production rate is 74.0 Tg (SOA) $a^{-1}$, which lies within the range of estimates from other global modelling studies but is substantially lower than top-down estimates. In addition, it is found that SOA concentrations, at least for the global mean, respond linearly to changes in reaction yields.

During NH summer, high biogenic and biomass burning emissions combine with enhanced levels of photochemistry, resulting in the global SOA production rate and global SOA burden also peaking during this season. Contrastingly, the net effect of two seasonal cycles (a winter peak in anthropogenic emissions and a summer peak in photochemistry that influences oxidation) results in a global anthropogenic SOA production rate which is constant all year around. However, the global anthropogenic SOA burden shows a seasonal cycle, peaking during NH spring and winter. This is due to the seasonal cycle of the SOA lifetime, which is shortest during summer. As peak anthropogenic SOA concentrations occur over India and China, the summertime reduction in anthropogenic SOA lifetime is possibly due to enhanced wet removal. Simulated annual average SOA concentrations from both biogenic and biomass burning sources peak in the SH, in tropical forest regions of South America and Africa. Contrastingly, simulated annual average SOA concentrations from anthropogenic sources peak in the NH, over industrialised and urban regions of India, China, Europe and USA.

The addition of new SOA sources also affects the concentrations of primary organic aerosol (POA) due to enhanced condensation-ageing. This increases the proportion of hydrophilic POA and, therefore, reduces the POA lifetime. POA

concentrations decrease over the Congo region and Siberia, which correspond to regions of increased SOA concentrations, available hydrophobic POA, and efficient wet removal.

Considering surface sites in the NH mid-latitudes, simulated SOA concentrations from the simulations with monoterpene as the only SOA source are substantially lower than observed (NMB = -91 %). Inclusion of all three new sources of SOA,
isoprene, anthropogenic and biomass burning, improves model agreement with observations (NMB = -50 %). This is primarily due to the inclusion of an anthropogenic source of SOA whereas inclusion of isoprene and biomass burning as sources of SOA have little effect on model agreement with observations. However, a substantial underestimate remains in simulated SOA concentrations. When the reaction yield of SOA formation from anthropogenic sources was increased from 13 to 40 % (production = 73.6 Tg a$^{-1}$), model agreement with observations improves even further (NMB = -10 %). However, simulated
SOA concentrations in urban environments are lower than observed (NMB = -34 %) whereas, further downwind (NMB = 14 %) and in remote environments (NMB = 12 %), simulated SOA concentrations are in relatively good agreement compared to observations albeit slightly overestimated. A large anthropogenic source of SOA may be plausible, however, the reaction yield required greatly exceeds that derived from most chamber studies to date, revealing a knowledge gap to be filled. Including POA oxidation may further improve the negative bias in modelled SOA and reduce POA lifetime, thus reducing the large
positive bias in simulated POA in remote regions.

However, these results relate primarily to the NH mid-latitudes where anthropogenic emissions are highest. Extending the observational dataset to include measurements in the SH, Santiago (Chile), Manaus (Brazil), and Welgegund (South Africa), shows that isoprene also has an effect on model agreement with observations, with simulated SOA concentrations closer to observed values at Welgegund (South Africa) but produce a greater overestimate at Manaus (Brazil) compared to observations.
However, the lack of observations in regions influenced by biogenic and biomass burning results in very little constraint on these sources of SOA. When considering aircraft campaigns, with monoterpene as the only source of SOA, simulated SOA concentrations are substantially lower than observed in all environments and at all altitudes. Inclusion of new sources of SOA improves model agreement further, again, primarily due to the inclusion of an anthropogenic source of SOA.

There are several limitations in this study. Firstly, as highlighted, the scarcity of available observations result in difficulty
in constraining simulated SOA. More observations of SOA are required in the SH, particularly over tropical forest regions of South America and Africa, where simulated SOA concentrations from biogenic and biomass burning sources are extremely high. The lumping of anthropogenic and biomass burning VOC species into single compounds also represents a significant uncertainty in this study. Emitted VOC species from each source are likely to have different emissions distributions, reaction kinetics and reaction yields which will likely result in differences in simulated SOA which will have not been captured in this
study. However, explicitly simulating each VOC is hindered by a lack of knowledge of the dominant species and the required computational expense. Additionally, in this study, using the UKCA model VOC oxidation products of low enough volatility to condense are lumped into a single surrogate compound (SOG). Therefore, this assumption does not account for the volatility distribution of oxidation products. Chemical ageing in the atmosphere may be more efficiently represented using the volatility basis set (VBS) (Donahue et al., 2006). Additionally, dry and wet deposition of SOA precursors is not included in this model

due to the lumping of species and uncertainties in deposition parameters. Including dry and wet deposition of SOA precursors will likely reduce SOA concentrations. Another limitation to this study is the absence of aqueous SOA formation in aerosols (Ervens, 2015) and cloud water (McNeill et al., 2012). Further laboratory studies are required to provide detailed oxidation mechanisms of VOC species such that they can be implemented into chemistry-climate models. Future modelling work will

evaluate dry deposition, wet deposition, and an evolving volatility distribution or SOA precursors, and their impacts on SOA formation.

Nevertheless, we have considered SOA formation from a number of different sources in a global composition-climate model, and compared against a consistent set of observations. In doing so, we have highlighted that, overall, the inclusion of new sources of SOA improves the ability of the UKCA model to simulate SOA distributions across many world regions.

Additionally, the new estimate of the global SOA budget from UKCA lies within the range of estimates from other global modelling studies. Future modelling work should aim to improve confidence in SOA formation mechanisms, and to explicitly simulate multigenerational oxidation products with evolving volatility. Furthermore, observations of SOA are required in regions influenced by biogenic and biomass burning emissions, such as South America and Africa.

*Acknowledgements.* This work is supported by Natural Environment Research Council (NERC; NE/L008947/1) and the Met Office through a CASE award. The development of UKCA and Fiona M. O'Connor are supported by the Joint UK BEIS/Defra Met Office Hadley Centre Climate Programme (GA01101). F.M. O'Connor also acknowledges additional funding received from the Horizon 2020 European Union's Framework Programme for Research and Innovation "Coordinated Research in Earth Systems and Climate: Experiments, Knowledge, Dissemination and Outreach (CRESCENDO)" project under grant

agreement no. 641816. Computer resources provided by the Met Office, the MONSooN supercomputer facility, were used for the UKCA simulations reported here. The MONSooN system is a collaborative facility supplied under the Joint Weather and Climate Research Programme (JWCRP), which is a strategic partnership between the Met Office and NERC. ERA-Interim data used in this study were provided by the European Centre for Medium Range Weather Forecasts (ECMWF).

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

**Table 1** – Reaction kinetics for SOA precursors in UKCA.

| Reaction | Rate coefficient (cm$^3$ s$^{-1}$) |
|---|---|
| $monoterpene + OH$ | $1.2 \times 10^{-11} \exp(444/T)$ |
| $monoterpene + O_3$ | $1.01 \times 10^{-15} \exp(-732/T)$ |
| $monoterpene + NO_3$ | $1.19 \times 10^{-12} \exp(490/T)$ |
| $isoprene + OH$ | $2.7 \times 10^{-11} \exp(390/T)$ |
| $isoprene + O_3$ | $1.0 \times 10^{-14} \exp(-1995/T)$ |
| $isoprene + NO_3$ | $3.15 \times 10^{-12} \exp(-450/T)$ |
| $VOC_{ANT} + OH$ | $1.2 \times 10^{-11} \exp(444/T)$ |
| $VOC_{BB} + OH$ | $1.2 \times 10^{-11} \exp(444/T)$ |

**Table 2** – Summary of simulations carried out in study. Reaction kinetics for each source are shown in Table 1. For the BBsSlow simulation, $VOC_{BB}$ assumes the reactivity of naphthalene (Atkinson and Arey, 2003).

| Simulation | SOA sources included | Reaction yield / % |
|---|---|---|
| Control | monoterpene | 13 |
| Iso | monoterpene | 13 |
|  | isoprene | 13 |
| BB | monoterpene | 13 |
|  | $VOC_{BB}$ | 13 |
| Ant | monoterpene | 13 |
|  | $VOC_{ANT}$ | 13 |
| AllSources | monoterpene | 13 |
|  | isoprene | 13 |
|  | $VOC_{BB}$ | 13 |
|  | $VOC_{ANT}$ | 13 |
| Iso (3%) | monoterpene | 13 |
|  | isoprene | 3 |
| Ant (40%) | monoterpene | 13 |
|  | $VOC_{ANT}$ | 40 |
| BB_slow* | monoterpene | 13 |
|  | $VOC_{BB}$ | 13 |

**Table 3** – Global annual SOA production from this study and the literature (Tg (SOA) a[-1]). In this study, estimates derived from the isoprene (Iso(3%)) and anthropogenic (Ant(40%)) sensitivity simulations in this study are indicated. All remaining estimates from this study are based on simulations using identical reaction yields of 13 %. From the literature, estimates derived from top-down and observation methods are indicated. The remaining estimates form the literature are based on bottom-up approaches.

| Component | SOA production / Tg (SOA) a[-1] | |
| --- | --- | --- |
| | This study | Literature |
| Biogenic | 23.9 [a] - 39.5 | 46.4 (Khan et al., 2017) |
| | | 26.8 (Henze et al., 2008) |
| | Monoterpene = 19.9 | 27.6 (Farina et al., 2010) |
| | | 18.6 (Tsigaridis and Kanakidou, 2007) |
| | Isoprene = 4[a] – 19.6 | 55 (Hoyle et al., 2007) |
| | | 14.9 (Henze and Seinfeld, 2006) |
| | | 97.5 (Hodzic et al., 2016) |
| | | 13 (Spracklen et al., 2011)[*] |
| Biomass burning | 9.5 | 15.5 (Hodzic et al., 2016) |
| | | 3-26 (Spracklen et al., 2011) [*] |
| | | 44-95 (Shrivastava et al., 2015) |
| | | 1-15 (Cubison et al., 2011)[+] |
| | | 34 (Hallquist et al., 2009)[*] |
| Anthropogenic | 24.6 – 70.0[b] | 1.6 (Farina et al., 2010) |
| | | 3.1 (Heald et al., 2011) |
| | | 19.2 (Hodzic et al., 2016) |
| | | 100 (Spracklen et al., 2011) [*] |
| Total | 48.5[a] – 74.0 – 119.4[b] | 132 (Hodzic et al., 2016) |
| | | 19 (13 – 121) (Tsigaridis et al., 2014) |
| | | 12-70 (Tsigaridis and Kanakidou, 2007) |
| | | 140 (50-380) (Spracklen et al., 2011) [*] |
| | | 250 (50 – 450) (Heald et al., 2010) [*] |
| | | 26.5 (Heald et al., 2011) |
| | | 280 – 1820 (Goldstein and Galbally, 2007) [*] |

[a] Estimated using an isoprene yield of 3 %

[b] Estimated using an anthropogenic yield of 40 %

[*] Estimated using top-down methods

**Table 4** – Summary of statistics for simulated SOA against observed oxygenated OA for simulations described in Table 2. NMB represents the normalised mean bias. Measurements cover period 2000-2010, model results are for the year 2000.

| | number of sites | Obs (AMS) mean ($\mu g\ m^{-3}$) | Model (UKCA) Control mean ($\mu g\ m^{-3}$) | Control NMB (%) | Iso mean ($\mu g\ m^{-3}$) | Iso NMB (%) | BB mean ($\mu g\ m^{-3}$) | BB NMB (%) | Ant mean ($\mu g\ m^{-3}$) | Ant NMB (%) | AllSources mean ($\mu g\ m^{-3}$) | AllSources NMB (%) | Ant(40%) mean ($\mu g\ m^{-3}$) | Ant(40%) NMB (%) | Iso(3%) mean ($\mu g\ m^{-3}$) | Iso(3%) NMB (%) |
|---|---|---|---|---|---|---|---|---|---|---|---|---|---|---|---|---|
| all sites | 37 | 3.68 | 0.33 | -91 | 0.59 | -84 | 0.38 | -90 | 1.30 | -65 | 1.83 | -50 | 3.32 | -10 | 0.39 | -89 |
| site type | | | | | | | | | | | | | | | | |
| urban | 14 | 4.76 | 0.36 | -93 | 0.64 | -86 | 0.42 | -91 | 1.28 | -53 | 1.92 | -60 | 3.17 | -34 | 0.42 | -91 |
| urban downwind | 6 | 3.93 | 0.59 | -85 | 1.07 | -73 | 0.63 | -84 | 1.85 | -53 | 2.55 | -35 | 4.50 | 14 | 0.70 | -82 |
| remote | 17 | 2.70 | 0.22 | -92 | 0.38 | -86 | 0.26 | -90 | 1.12 | -58 | 1.50 | -44 | 3.02 | 12 | 0.26 | -90 |
| Continent | | | | | | | | | | | | | | | | |
| Europe | 13 | 4.20 | 0.15 | -93 | 0.24 | -88 | 0.16 | -92 | 0.84 | -58 | 0.97 | -52 | 2.25 | 12 | 0.18 | -91 |
| North America | 12 | 2.02 | 0.70 | -83 | 1.26 | -70 | 0.79 | -81 | 1.68 | -60 | 2.64 | -37 | 3.72 | -12 | 0.83 | -80 |
| Asia | 12 | 4.77 | 0.11 | -98 | 0.22 | -95 | 0.16 | -97 | 1.35 | -72 | 1.81 | -62 | 3.94 | -17 | 0.14 | -97 |

**Table 5** – Summary of statistics for simulated POA against observed hydrocarbon-like OA for simulations described in Table 2. NMB indicated normalised mean bias. Measurements cover period 2000-2010, model results are for the year 2000.

| | number of sites | Obs (AMS) mean ($\mu g\ m^{-3}$) | Model (UKCA) Control mean ($\mu g\ m^{-3}$) | Control NMB (%) | Iso mean ($\mu g\ m^{-3}$) | Iso NMB (%) | BB mean ($\mu g\ m^{-3}$) | BB NMB (%) | Ant mean ($\mu g\ m^{-3}$) | Ant NMB (%) | AllSources mean ($\mu g\ m^{-3}$) | AllSources NMB (%) | Ant(40%) mean ($\mu g\ m^{-3}$) | Ant(40%) NMB (%) | Iso(3%) mean ($\mu g\ m^{-3}$) | Iso(3%) NMB (%) |
|---|---|---|---|---|---|---|---|---|---|---|---|---|---|---|---|---|
| all sites | 37 | 1.25 | 0.71 | -43 | 0.70 | -43 | 0.69 | -45 | 0.69 | -44 | 0.69 | -45 | 0.68 | -45 | 0.72 | -41 |
| site type | | | | | | | | | | | | | | | | |
| urban | 14 | 2.79 | 0.84 | -70 | 0.82 | -71 | 0.82 | -71 | 0.82 | -70 | 0.82 | -71 | 0.82 | -71 | 0.84 | -70 |
| urban downwind | 6 | 0.78 | 0.45 | -41 | 0.46 | -41 | 0.45 | -42 | 0.45 | -42 | 0.45 | -42 | 0.45 | -42 | 0.46 | -40 |
| remote | 17 | 0.14 | 0.70 | 410 | 0.69 | 400 | 0.67 | 391 | 0.67 | 384 | 0.66 | 375 | 0.65 | 373 | 0.70 | 409 |
| Continent | | | | | | | | | | | | | | | | |
| Europe | 13 | 0.77 | 0.32 | -59 | 0.31 | -59 | 0.31 | -59 | 0.31 | -59 | 0.31 | -60 | 0.31 | -60 | 0.32 | -58 |
| North America | 12 | 1.65 | 0.58 | -65 | 0.56 | -66 | 0.55 | -67 | 0.56 | -66 | 0.56 | -66 | 0.56 | -66 | 0.58 | -65 |
| Asia | 12 | 1.28 | 1.26 | -2 | 1.25 | -2 | 1.22 | -5 | 1.22 | -5 | 1.20 | -6 | 1.20 | -7 | 1.27 | -1 |

**Table 6 –** Summary of statistics for simulated OA against observed OA for simulations described in Table 2. NMB indicated normalised mean bias. Measurements cover period 2000-2010, model results are for the year 2000.

| | number of sites | Obs (AMS) mean (µg m⁻³) | Model (UKCA) | | | | | | | | | | | | | |
|---|---|---|---|---|---|---|---|---|---|---|---|---|---|---|---|---|
| | | | Control | | Iso | | BB | | Ant | | AllSources | | Ant(40%) | | Iso(3%) | |
| | | | mean (µg m⁻³) | NMB (%) | mean (µg m⁻³) | NMB (%) | mean (µg m⁻³) | NMB (%) | mean (µg m⁻³) | NMB (%) | mean (µg m⁻³) | NMB (%) | mean (µg m⁻³) | NMB (%) | mean (µg m⁻³) | NMB (%) |
| all sites | 37 | 4.95 | 1.05 | -79 | 1.30 | -74 | 1.07 | -78 | 1.99 | -60 | 2.51 | -49 | 4.00 | -19 | 1.11 | -78 |
| site type | | | | | | | | | | | | | | | | |
| urban | 14 | 7.62 | 1.20 | -84 | 1.47 | -81 | 1.23 | -84 | 2.10 | -72 | 2.74 | -64 | 3.98 | -47 | 1.26 | -83 |
| urban downwind | 6 | 4.72 | 1.05 | -78 | 1.52 | -69 | 1.08 | -77 | 2.30 | -51 | 3.01 | -36 | 4.95 | -5 | 1.65 | -75 |
| remote | 17 | 2.83 | 0.92 | -67 | 1.07 | -62 | 0.93 | -67 | 1.79 | -37 | 2.15 | -24 | 3.68 | 30 | 0.97 | -66 |
| Continent | | | | | | | | | | | | | | | | |
| Europe | 13 | 2.78 | 0.47 | -83 | 0.55 | -80 | 0.47 | -83 | 1.15 | -58 | 1.28 | -54 | 2.56 | -7 | 0.50 | -82 |
| North America | 12 | 5.92 | 1.28 | -78 | 1.82 | -69 | 1.34 | -78 | 2.24 | -62 | 3.20 | -46 | 4.27 | -28 | 1.41 | -76 |
| Asia | 12 | 6.05 | 1.38 | -77 | 1.47 | -76 | 1.37 | -77 | 2.56 | -58 | 3.00 | -50 | 5.14 | -15 | 1.41 | -77 |

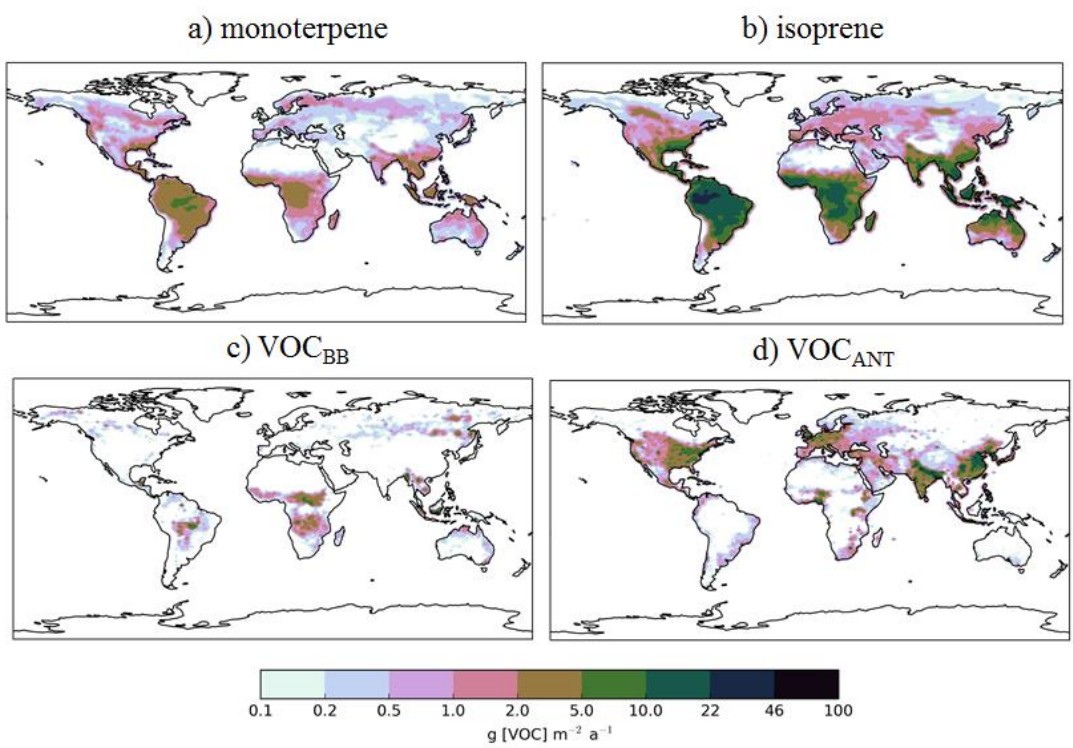

**Figure 1:** Annual-total SOA precursor emissions from the different global VOC sources; monoterpene and isoprene taken from Guenther et al. (1995), VOC$_{BB}$ taken from EDGAR, and VOC$_{ANT}$ taken from Lamarque et al. (2010) Units are g (VOC) m$^{-2}$ a$^{-1}$.

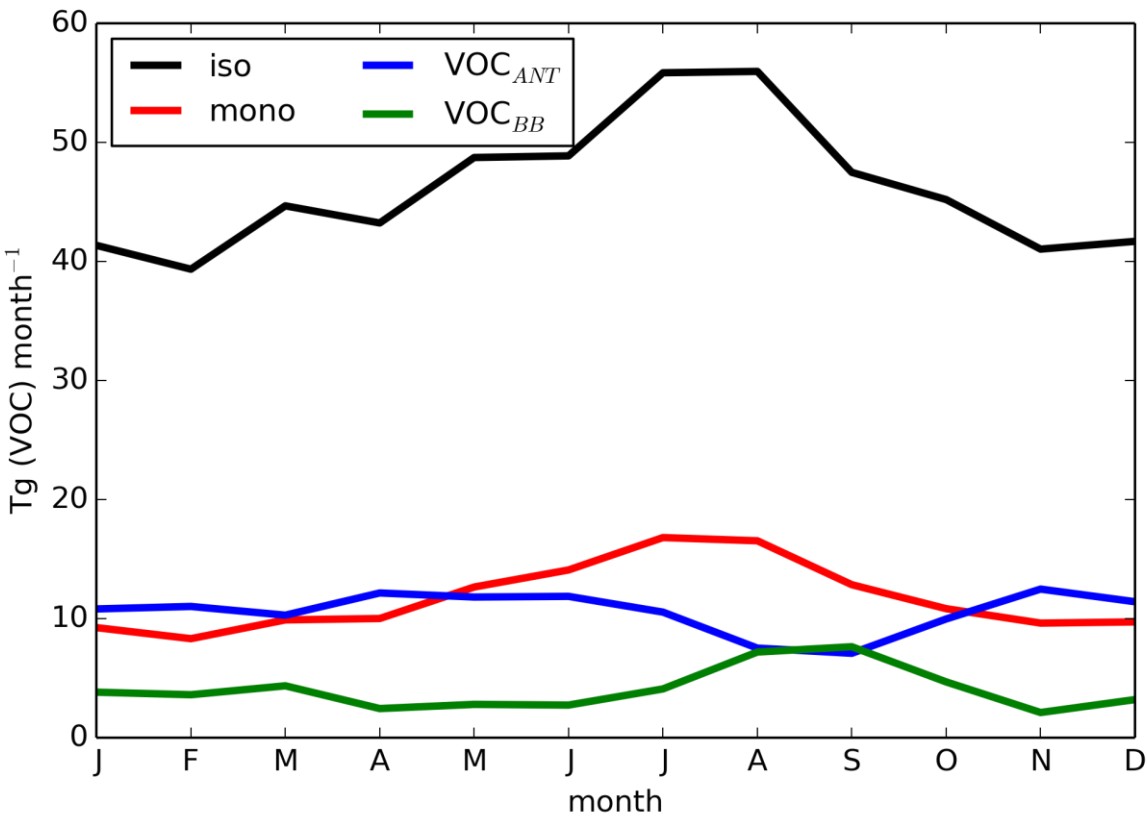

**Figure 2** – Seasonality of global SOA precursor emissions from the different VOC sources: monoterpene (red), isoprene (black), VOC_ANT (blue) and VOC_BB (green) (Tg (VOC) month[-1]).

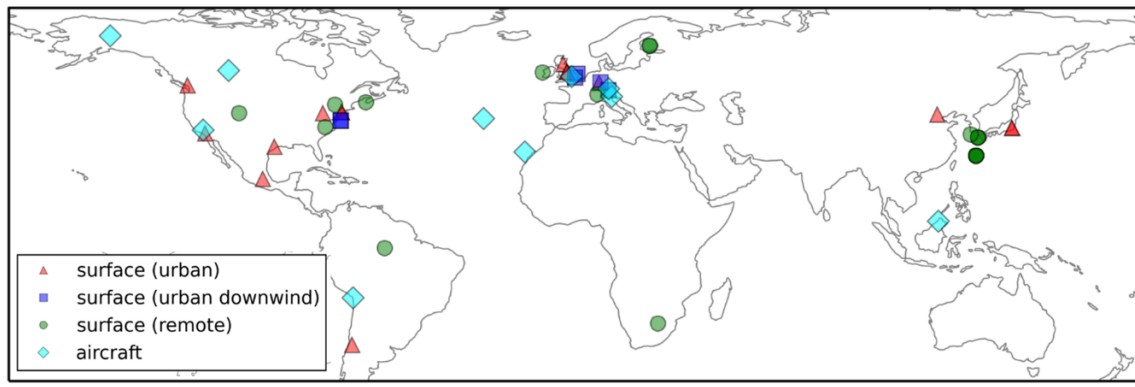

**Figure 3** – Global map showing the 40 surface AMS observations, originally compiled Zhang et al. (2007) classified as urban (red triangles), urban downwind (blue squares) or remote (green circles). Of the surface observations, 37 have been classified as hydrocarbon-like OA and oxygenated-OA. Observations from 10 aircraft campaigns, originally compiled by Heald et al. (2011), are also shown (light blue diamonds). These remain as total OA.

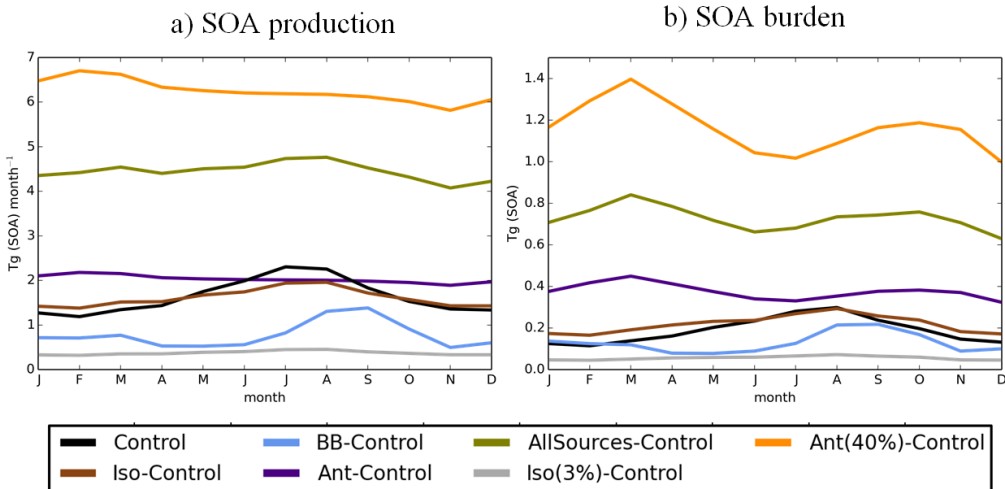

**Figure 4**– Monthly average global SOA (a) production (Tg (SOA) month$^{-1}$) and (c) burden (Tg (SOA)), simulated by UKCA for the control simulation in black. For the other UKCA simulations described in Table 2, the monthly average global SOA production and burden are shown relative to the control simulation.

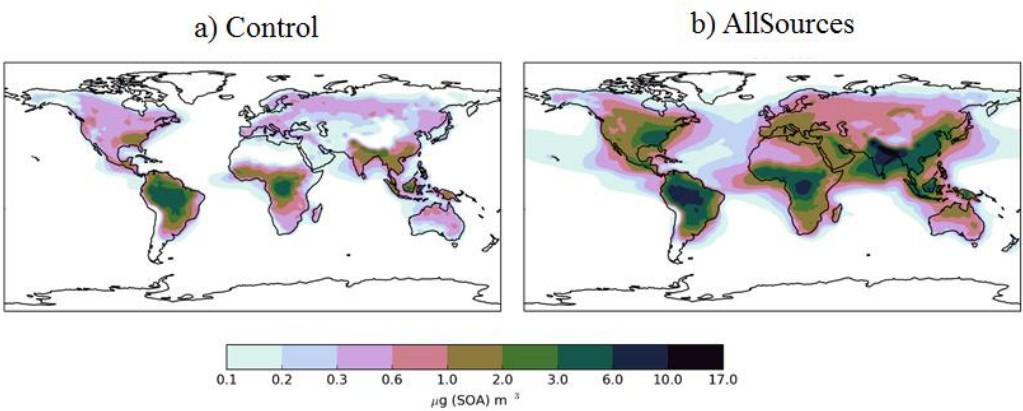

**Figure 5** – Annual-average surface SOA concentrations (µg m⁻³) for Control (monoterpene) and AllSources (monoterpene, isoprene, VOC_BB, VOC_ANT) simulations described in Table 2.

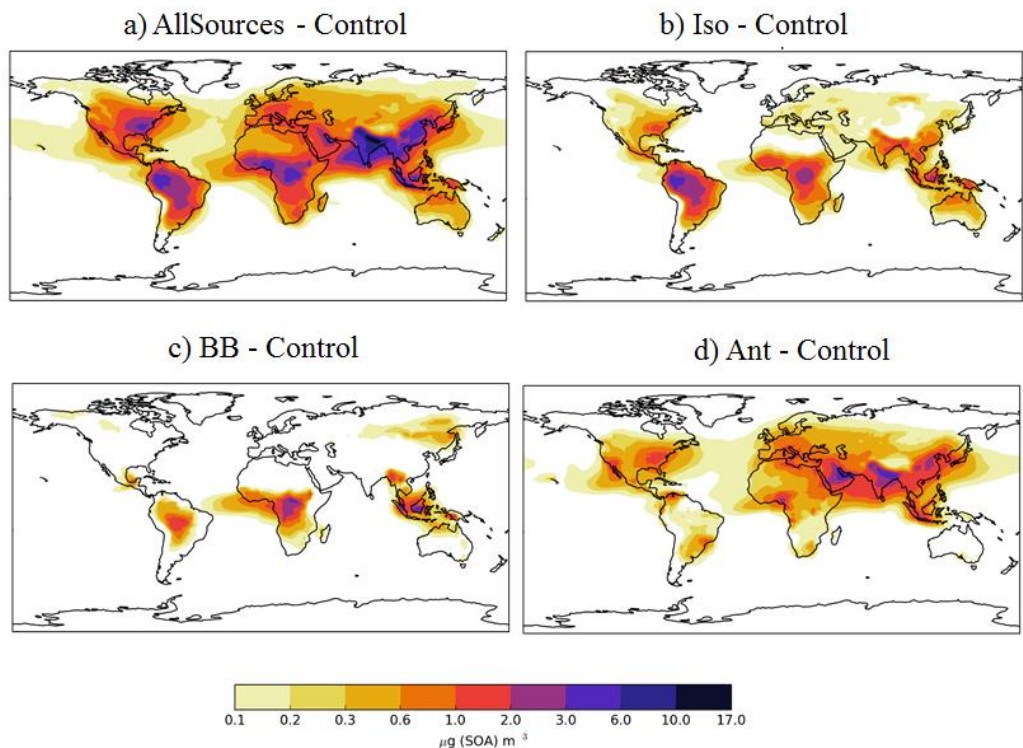

5    **Figure 6** – Differences in annual-average SOA concentrations (µgm$^{-3}$) relative to the control run for simulations (a) AllSources, (b) Iso, (c) BB and (d) Ant Simulations are described in Table 2.

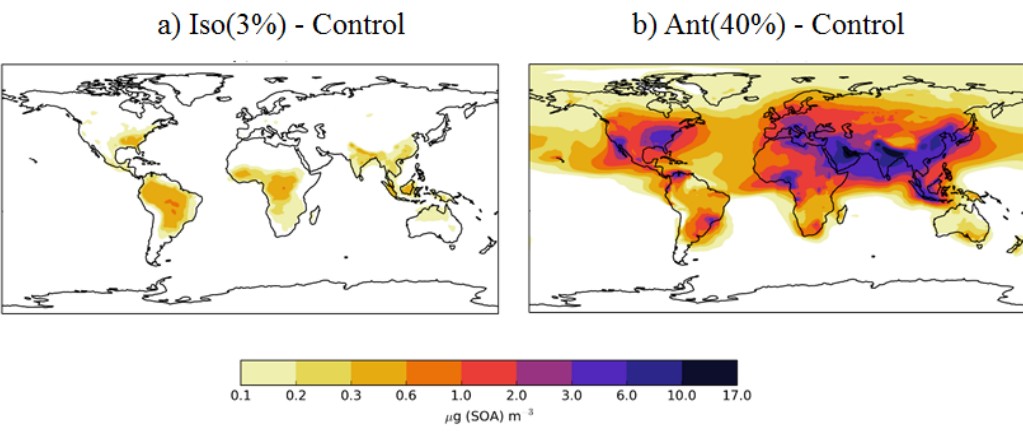

**Figure 7 -** Differences in annual-average SOA concentrations (μgm⁻³) relative to the control run for further sensitivity simulations (a) Ant (40%), (b) Iso (3%). Simulations are described in Table 2.

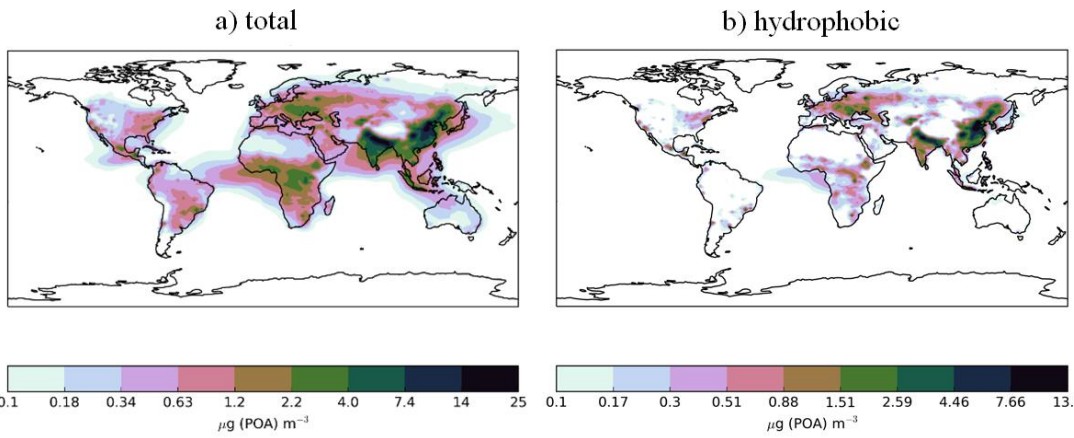

**Figure 8** – Annual-average surface (a) total (hydrophilic and hydrophobic), and (b) hydrophobic only, POA concentrations (µgm$^{-3}$) for the Control simulation. Within UKCA, all POA is emitted into the hydrophobic modes and re-distributed into the hydrophilic modes through condensation-ageing.

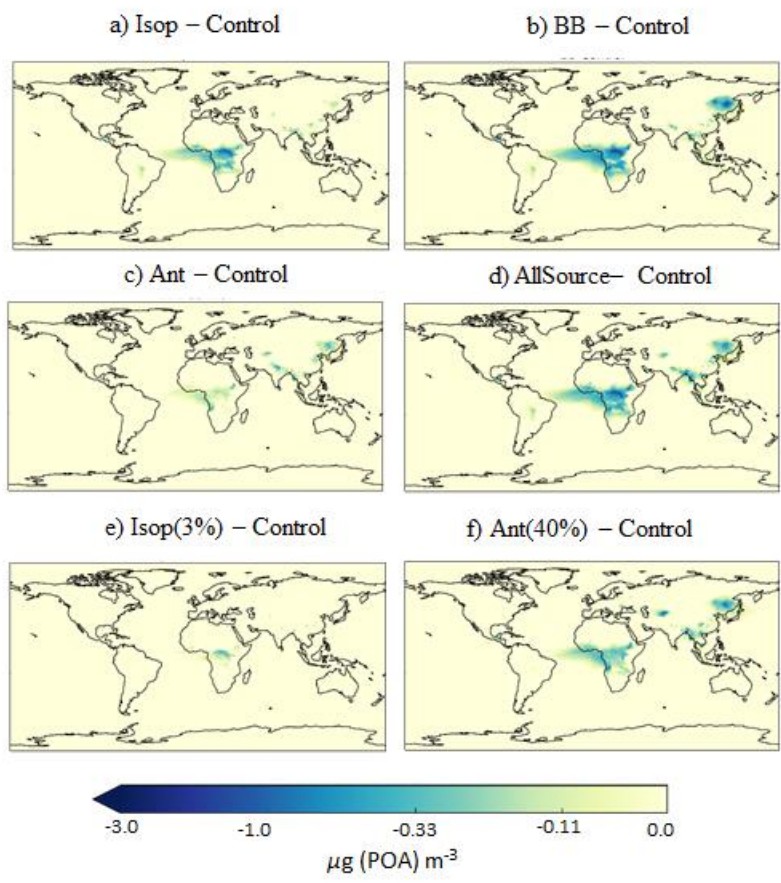

**Figure 9** – Differences in annual average surface total POA concentrations (µgm⁻³) relative to the control run for simulations (a) AllSources, (b) Iso, (c) BB and (d) Ant, which are described in Table 2. Regions of decreased POA correspond to regions of increased SOA concentrations, availability of hydrophobic POA and efficient wet removal.

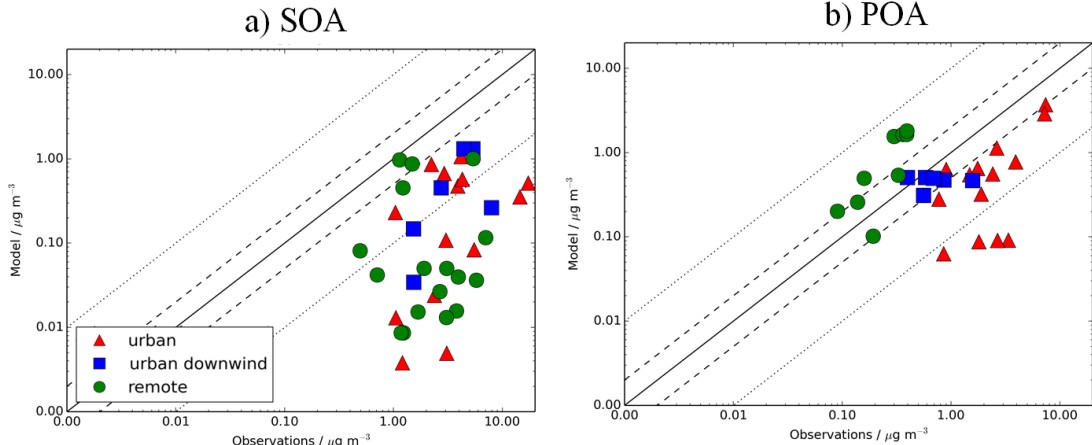

**Figure 10** – Simulated versus observed (a) SOA and (b) POA concentrations (µg m⁻³). Observed oxygenated-OA is assumed to be comparable to simulated SOA, whereas observed hydrocarbon-like OA is assumed to be comparable to simulated POA. Simulated concentrations are taken from the control run for the year 2000, described in Table 2. Observations for the time period 2000-2010 are classified as urban (red triangles), urban downwind (blue squares) or remote (green circles). The 1:1 (solid), 1:2 and 2:1 (dashed), and 1:10 and 10:1 (dotted) lines are indicated. Model-observation statistics for SOA, POA and OA are shown in Tables 4, 5 and 6, respectively.

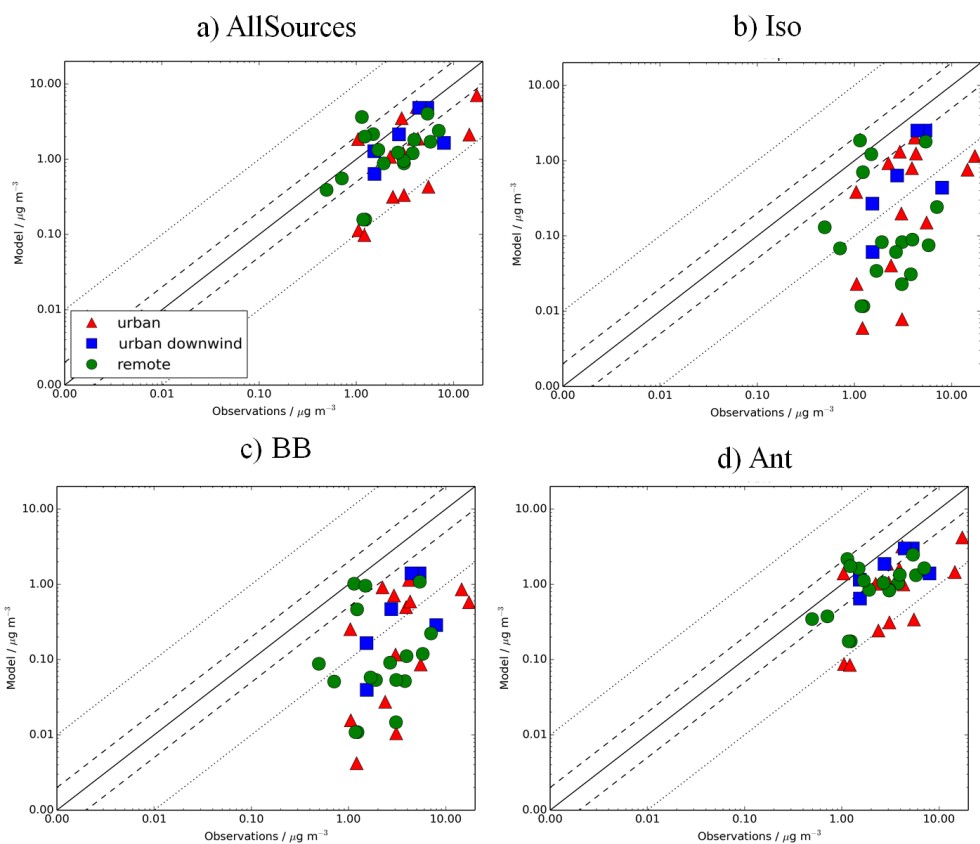

**Figure 11 –** Simulated versus observed SOA (µg m⁻³) for simulations (a) AllSources, (b) Iso, (c) BB, and (d) Ant, described in Table 2. Observations for the time period 2000-2010 are classified as urban (red triangles), urban downwind (blue squares) or remote (green circles). Observed oxygenated-OA is assumed to be comparable to simulated SOA. The 1:1 (solid), 1:2 and 2:1 (dashed), and 1:10 and 10:1 (dotted) lines are indicated. Model-observation statistics for SOA are shown in Table 4.

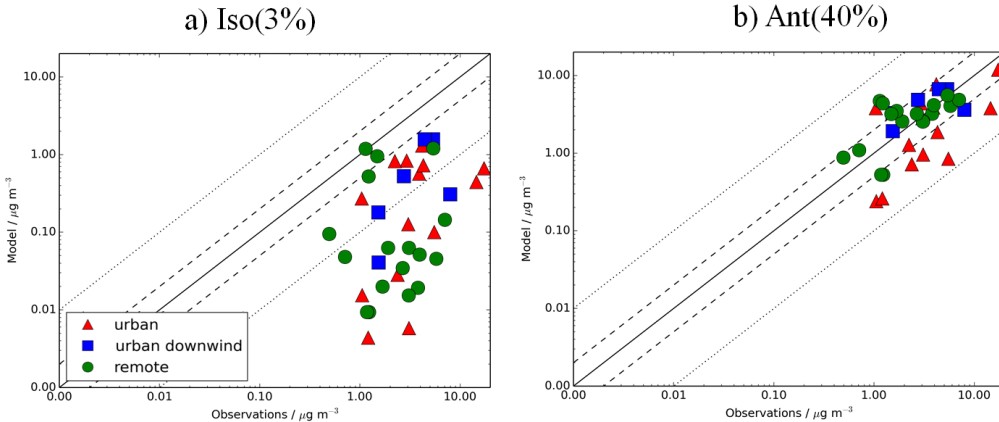

**Figure 12** – Simulated versus observed SOA (µg m⁻³) for sensitivity simulations (a) Iso (3%) and (b) Ant (40%), described in Table 2. Observations for the time period 2000-2010 are classified as urban (red triangles), urban downwind (blue squares) or remote (green circles). Observed oxygenated-OA is assumed to be comparable to simulated SOA. The 1:1 (solid), 1:2 and 2:1 (dashed), and 1:10 and 10:1 (dotted) lines are indicated. Model-observation statistics for SOA are shown in Table 4.

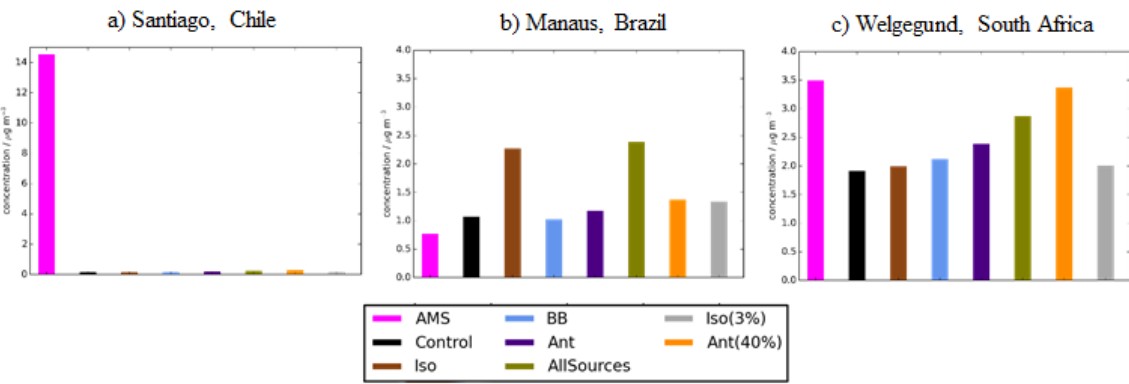

**Figure 13**- Simulated and observed OA surface concentrations (μg m$^{-3}$) over an urban environment, (a) Santiago (Chile) and remote environments, (b) Manaus (Brazil) and (c) Welgegund (South Africa). Bars indicate observed (pink) and simulated OA surface concentrations from the control (black), Iso (brown), BB (light blue), Ant (dark blue), AllSources (green), Ant (40%) (yellow) and Iso (3%) (grey). Model simulations are described in Table 2.

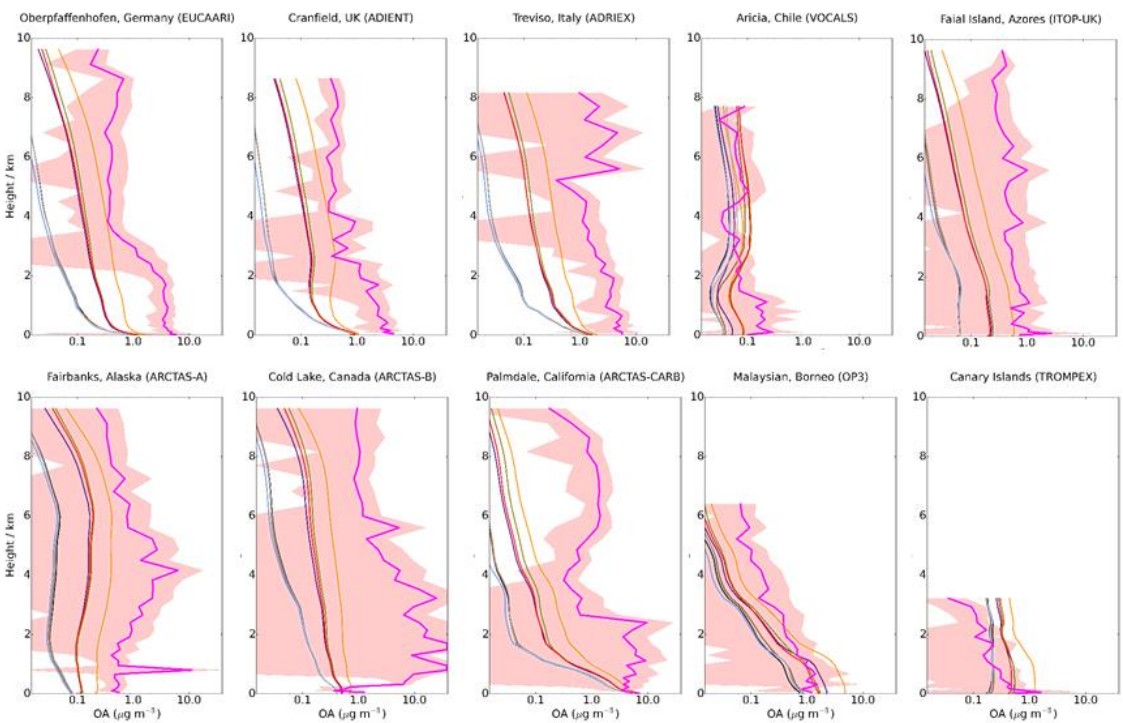

**Figure 14** – Mean vertical profile of OA (µg m$^{-3}$) for 10 field campaigns with the mean UKCA for the simulations described in Table 2. The standard deviation of the binned observations at each model layer are shown (peach envelope). Colours used are identical to Figure 13.

