# Peer review of "The impact of biogenic, anthropogenic and biomass burning volatile organic compound emissions on regional and seasonal variations in secondary organic aerosol"

_Atmospheric Chemistry and Physics, 2017_

## Referee Comment (RC1) · Anonymous Referee #1 · 7 Dec 2017

This paper is a global modeling study of SOA. The paper is generally well written. I liked their Introduction. However, there are several points that need to be clarified before publication. Following are my major comments:

Page 10 near the top: Why does anthropogenic SOA have longer lifetime than biogenic and biomass burning SOA?

Biomass burning: Did the authors consider high altitude emissions of biomass burning VOCs? These can increase tropospheric long range transport of BBSOA and increase its lifetime.

Biomass burning SIVOC are not considered and biomass burning SOA formation rate

is assumed to be similar to monoterpenes. There are a several issues with this. First, monoterpenes are not a major fraction of biomass burning SIVOC/VOC emissions, although I agree they are also emitted during wildfire burns. Most of the biomass burning SIVOCs would be branched/cyclic akanes or other long-chain carbon compounds. Second, monoterpenes react much faster than SIVOC . The emissions of VOCBB and 13% yield are also arbitrary. The authors need to justify their choices here and cite appropriate measurements of biomass burning emissions of VOC/SIVOC, and their choice of yields.

For example, can model-measurement comparisons be used to understand if biomass burning SOA formation is fast (similar to monoterpenes used in this study) or is slow (as given by SIVOCs that are slower reacting but not considered here)?

Although SIVOC chemistry is uncertain, measurements certainly show evidence for large amounts of missing SIVOC emissions from biomass burning (see Yokelson et al. 2013 and related discussions in the global modeling study of Shrivastava et al. 2015). This means SIVOC emissions and chemistry cannot be completely neglected, they just need better constraints.

In summary, what is the justification for neglecting SIVOC emissions/chemistry in this paper?

Page 16: Welgegund: Can biomass burning be a large missing SOA source at this cite rather than anthropogenic SOA? For example, Shrivastava et al. 2015 report that including biomass burning as SOA source improves model-measurement agreement in terms of seasonality of SOA at this site.

Figure 14: The various lines need captions. The text and the legends in this figure need to be made clearer for readability. For example, it's not clear which is monoterpene only versus total all source SOA simulation.

The authors report that inclusion of biomass burning SOA source does not improve

model performance with respect to aircraft measurements especially during ARCTAS. In sharp contrast, Shrivastava et al. 2015 reported a large increase in model performance especially at high altitudes, with respect to ARCTAS field campaign when they included biomass burning SOA source. Clearly, this reflects the large difference in biomass burning SOA treatment between this study and Shrivastava et al. 2015 study. Some discussions about why the authors don't see an improvement due to biomass burning SOA is warranted and also how their SOA treatment for biomass differs from Shrivastava et al. 2015 study.

---

## Referee Comment (RC2) · Anonymous Referee #2 · 18 Feb 2018

Kelly et al. update a global chemistry-climate model (UKCA) for new sources of secondary organic aerosol (SOA) and report on model predictions and the model-measurement comparison from this updated model. They find that, on average, the inclusion of new sources of SOA improves model performance against organic aerosol (OA) mass concentrations, POA-SOA splits, and OA vertical profiles but caution that the model still does not include some major SOA formation pathways and processes (e.g., varying volatility, aqueous chemistry) and remains unconstrained in the southern hemisphere due to a paucity of observations.

Kelly et al. have done an excellent job of reporting results from the model simulations

and the model-measurement comparison. The manuscript is also very well written and makes for very easy reading. The findings and discussion from this work will be helpful to the community. However, I found that the methods used did not reflect the gaps/uncertainties discussed in the introduction nor come close to the current state-of-the-science for treating SOA formation pathways and processes in atmospheric models. While novel for the UKCA, the sources/formation pathways explored in this work are routine for some of the other global chemistry-climate models and I was struggling to see how this work was novel and offered fresh insights into the SOA budget that haven't been explored in earlier work. This work definitely needs to be published but I am not comfortable recommending publication in Atmospheric Chemistry and Physics given the lack of novelty.

I have listed just a few of my concerns related to the methods:

1. The POA and SOA need to be treated as semi-volatile and reactive to better model the OA mass concentrations and its sensitivity to chemistry and changes in temperature. One example where this would influence one of the findings reported in this work is that SOA monolayers coated on POA could evaporate with dilution/chemistry and transition POA back into its hydrophobic mode. Another complication related to treating the semi-volatile nature of OA that has surfaced recently is if organic particles achieves instantaneous equilibrium with the organic vapors and how the phase state (i.e., diffusion limitations within the particle) might influence the timescales to achieve equilibrium. A semi-volatile treatment should be trivial to include with say a 2-product model.

2. There is plenty of evidence that the chemical lifetimes and SOA mass yields are very different for different SOA precursors (in addition to being a function of the OA mass loading), which can be very easily reflected in this work (regardless of whether the OA is treated as semi-volatile or non-volatile). Furthermore, SOA mass yields need to be corrected for vapor wall loss artifacts experienced in chamber experiments.

[Figure]

3. Emissions of semi-volatile and intermediate-volatility organic compounds contribute significantly to anthropogenic SOA precursors (and possibly even biogenic SOA precursors) and need to be explicitly modeled. These might help reduce the under predictions in urban areas and elsewhere. On a related note, it is unclear to me how the model-measurement comparison at urban locations needs to be evaluated. What fraction of the under-prediction can be seemingly attributed to the model resolution?

4. Globally, aqueous processing of organic compounds in aerosol water and clouds is probably a very important source/sink of OA and needs to be included. At the very least, one needs to consider IEPOX, glyoxal, and in-cloud formation of a few dominant organic acids in the model.

---

## Author Comment (AC1) · 27 Mar 2018

**We thank both reviewers for their insightful feedback on this study which has considerably improved the manuscript. For each of the reviewers' comments (reiterated here in italics), we have provided a response and, we have also provided the modified text within the updated manuscript (indicated by quotation marks). In our revised manuscript, modified text is highlighted using tracked changes.**

**Referee #1**

1. *This paper is a global modeling study of SOA. The paper is generally well written. I liked their Introduction. However, there are several points that need to be clarified before publication. Following are my major comments*

- We thank the reviewer for their positive feedback.

2. *Page 10 near the top: Why does anthropogenic SOA have longer lifetime than biogenic and biomass burning SOA?*

- This is a good point that deserves further discussion in our manuscript. Our understanding is that the difference in lifetimes between the various SOA components is due to differences in the horizontal spatial distributions of precursor emissions and precipitation patterns. We have expanded on our discussion of SOA lifetime in the model simulation descriptions and results (sections 2.3 and 4.1 respectively) of the revised manuscript.

- Page 9 line 12:
  "For all SOA components and across all simulations, SOA is solely removed by wet and dry deposition."

  Page 12 lines 5–16:
- "The variation in lifetime for the different SOA components is likely due to differences in the spatial distributions of SOA precursor emissions, as well as the extent of co-location of emissions and precipitation.  Biogenic and biomass burning VOCs, primarily located in tropical forest regions of the southern hemisphere, experience different precipitation compared to anthropogenic VOCs, which are primarily released in urban and industrial regions of the northern hemisphere. Vertical gradients in SOA production can also affect the SOA lifetime. However, in this study, all SOA precursors are emitted at the surface, hence, the various SOA components in this study likely have very similar vertical gradients. Shrivastava et al. (2105) find that the SOA lifetime substantially increases when biomass burning precursors are emitted at higher altitudes, where SOA is less susceptible to deposition.  The range in SOA lifetimes over the different simulations in this study is in agreement with Tsigaridis et al. (2014), which ranged from 2.4 – 15 days.

These SOA lifetimes are also in good agreement with Hodzic et al. (2016) who estimate the SOA lifetime from biogenic VOCs, anthropogenic and biomass burning VOCs combined, and anthropogenic and biomass burning S/IVOCs are 2.2, 3.3 and 3.0 days, respectively."

3. *Biomass burning: Did the authors consider high altitude emissions of biomass burning VOCs? These can increase tropospheric long range transport of BBSOA and increase its lifetime.*

- We did not consider high altitude $VOC_{BB}$ emissions. We agree that higher altitude emissions would increase the SOA burden and its lifetime, and have modified the discussion on global SOA budget in Section 4.1 to make this clearer:

  Page 12 line 9:
  "Vertical gradients in SOA production can also affect the SOA lifetime. However, in this study, all SOA precursors are emitted at the surface. Shrivastava et al. (2105) find that the SOA lifetime substantially increases when biomass burning precursors are emitted at higher altitudes. "

4. *Biomass burning SIVOC are not considered and biomass burning SOA formation rate is assumed to be similar to monoterpenes. There are a several issues with this. First, monoterpenes are not a major fraction of biomass burning SIVOC/VOC emissions, although I agree they are also emitted during wildfire burns. Most of the biomass burning SIVOCs would be branched/cyclic akanes or other long-chain carbon compounds. Second, monoterpenes react much faster than SIVOC.*

- We thank the reviewer for this comment and we acknowledge there are several challenges when treating SOA production from biomass burning VOCs ($VOC_{BB}$). Firstly, we agree, monoterpenes are not the major fraction of $VOC_{BB}$ emissions. Our $VOC_{BB}$ is a lumped species, representing a mixture of organic compounds. As the dominant $VOC_{BB}$ leading to SOA production is unknown, selecting a rate constant for $VOC_{BB}$ is extremely challenging. Therefore, we initially chose to assume that $VOC_{BB}$ has a similar reactivity to monoterpene. However, we acknowledge that $VOC_{BB}$ may not be as reactive as monoterpene. In light of this, we have performed an additional sensitivity simulation where $VOC_{BB}$ assumes the reactivity of naphthalene, which has been used as a surrogate compound for IVOCs (Pye and Seinfeld, 2010) and is roughly 50 % less reactive than monoterpene. We have added this to the revised manuscript in the methods (section 2.3 including Table 2) and results (section 4.1).

Page 9 line 11:
"Eight model simulations were performed using…"

Page 9 lines 20-23:
"A final sensitivity study tests the influence of the assumed reactivity of $VOC_{BB}$ on SOA. Here, with a reaction yield of 13 %, the reactivity is assumed to be identical to naphthalene. Naphthalene was chosen as it has been used as a surrogate compound to represent IVOCs in a global modelling study (Pye and Seinfeld, 2010), and is roughly 50 % less reactive than monoterpene (Atkinson and Arey, 2003)."

Table 2 now includes this additional sensitivity simulation.

Page 14 line 34 – page 15 line 7:
"The influence of assumed reactivity on simulated SOA from biomass burning was investigated in an additional sensitivity simulation, where $VOC_{BB}$ adopts the reactivity of naphthalene, an aromatic species (Table 2; section 2.3) which has been used as a surrogate compound for IVOCs (Pye and Seinfeld, 2010). Compared to monoterpene, naphthalene is roughly 50 % less reactive (Atkinson and Arey, 2003). However, despite this substantial reduction in reactivity, the global annual-total SOA production rate from biomass burning VOCs is reduces by less than 1 %. Also, the simulated spatial distributions are almost identical for the two $VOC_{BB}$ species. Like all other SOA precursors in this study, $VOC_{BB}$ does not undergo dry or wet deposition. Therefore, a reduction in the reactivity of $VOC_{BB}$ does not affect the fate of this compound."

*The emissions of VOCBB and 13% yield are also arbitrary. The authors need to justify their choices here and cite appropriate measurements of biomass burning emissions of VOC/SIVOC, and their choice of yields. For example, can model-measurement comparisons be used to understand if biomass burning SOA formation is fast (similar to monoterpenes used in this study) or is slow (as given by SIVOCs that are slower reacting but not considered here)?*

-   We do not feel that the $VOC_{BB}$ emissions we use are arbitrary as we describe in the manuscript, how BB emissions of CO are used to define the spatial distribution and seasonal cycle of $VOC_{BB}$, and that the total emission rate for $VOC_{BB}$ is then scaled to equal non-methane VOC emissions from biomass burning, estimated by EDGAR. We acknowledge that we did not explain the assumption of a 13% yield for SOA from $VOC_{BB}$ clearly. In the default version of the model, monoterpene is assumed to have an SOA yield of 13 %. As new sources were added, we chose to initially assume identical reaction yields. In addition the assumption of identical reaction yields for each VOC source type allows us to attribute differences in SOA concentrations solely to differences

in the spatial pattern, seasonality, and magnitude of VOC precursor emissions.  As outlined above, we have conducted an additional sensitivity simulation to explore the effect of assumed reaction kinetics on SOA and added these results to the revised paper. We have also added additional text to section 4.2 to a) expand our discussion in the introduction (section 1)  on the highly uncertain laboratory-derived reaction yields, and b) in the methods (section 2.2) outline the difficulty in selecting reaction yields for surrogate compounds in the model (for both $VOC_{BB}$ and $VOC_{ANT}$).

Page 5 lines 10-19:
"The SOA reaction yields, which are typically determined in environmental chambers, are highly uncertain. Firstly, the VOC sources implemented into models often represent a mixture of different compounds, which may have differing reaction yields. For example, a recent environmental chamber study found that the SOA reaction yield varies from 2.5 to 16.9 % depending on the particular monoterpene species (Zhao et al., 2015). Secondly, reaction yields vary substantially from one study to another. Considering aromatic compounds, which are typically associated with SOA production from anthropogenic and biomass burning, estimates of SOA reaction yields from aromatic compounds from different chamber studies range from 1 to 73 (Odum et al., 1996;Odum et al., 1997;Song et al., 2005;Ng et al., 2007b;Chan et al., 2009). Part of this uncertainty is due to differences in environmental chamber conditions. For aromatic compounds, factors such as relative humidity (Hinks et al., 2018) and $NO_x$ (Ng et al., 2007a) influence oxidation pathways which, in turn, control the yields of SOA."

Page 8 lines 14-34:
"The anthropogenic and biomass burning VOCs are lumped species, which results in difficulty in selecting reaction kinetics for these species. For all simulations, $VOC_{ANT}$ and $VOC_{BB}$ are assumed to react solely with OH. Initially, $VOC_{ANT}$ and $VOC_{BB}$ are assumed to have identical reactivity to monoterpene. As monoterpene is a relatively reactive species, this provides an upper estimate for the rate for anthropogenic and biomass burning VOC oxidation. A lower estimate of SOA production from $VOC_{BB}$ is provided by assuming reactivity of naphthalene. Naphthalene has been used as surrogate compound for  IVOCs (Pye and Seinfeld, 2010) and is roughly 50 % less reactive than monoterpene. Both monoterpene and naphthalene species are used to represent the reactivity of $VOC_{ANT}$/$VOC_{BB}$ as they provide relatively wide estimates of the reactivity of these surrogate compounds. For all the new species added to SOA production: isoprene, $VOC_{ANT}$ and $VOC_{BB}$, initially, a reaction yield of 13 % is applied. As discussed in Section 1, reaction yields vary from one study to another, as well as within individual studies. Furthermore, $VOC_{ANT}$ and $VOC_{BB}$ are surrogate compounds, representing a mixture of species and, therefore, preventing selection of molecule-specific reaction yields from laboratory studies. The initial assumption of identical reaction yields for all species does not negate the findings from laboratory

studies, which suggest reaction yields are highly dependent on molecular structure. However, the substantial uncertainties of reaction yields, coupled with these species representing lumped species, preventing accurate selection of laboratory-derived reaction yields for specific compounds. In addition, identical reaction yields allows differences in SOA concentrations to be solely attributed to differences in the spatial pattern, seasonality, and magnitude of VOC precursor emissions. However, the influence of accounting for differences in reaction yields is explored in two additional simulations described below; the reaction yields for isoprene is assumed to be 3 %, which is suggested by (Kroll et al., 2005, 2006). Also, the reaction yield for $VOC_{ANT}$ is increased from 13 to 40 %, which is motivated by the widespread model negative bias in urban environments among global modelling studies."

5. *Although SIVOC chemistry is uncertain, measurements certainly show evidence for large amounts of missing SIVOC emissions from biomass burning (see Yokelson et al. 2013 and related discussions in the global modeling study of Shrivastava et al. 2015).This means SIVOC emissions and chemistry cannot be completely neglected, they just need better constraints. In summary, what is the justification for neglecting SIVOC emissions/chemistry in thispaper?*

We thank the reviewer for this comment as this has highlighted to us that the objective of our study was not clearly presented. Firstly, we acknowledge that S/IVOCs may be an important source of SOA several times within the original manuscript (page 4 line 6 and page 14 line 26 of original manuscript). However, the objective of our study was to quantify the role of the three main VOC source types (biogenic, anthropogenic and biomass burning) on the global SOA budget and model agreement with observations. To clarify, our objective was not to evaluate a state-of-the-art treatment of SOA, which would require VOCs, S/IVOCs and aqueous production. In this study, we have focussed on one SOA source: VOCs, and covered its emissions types. We have now updated the title of the manuscript to:
"The impact of biogenic, anthropogenic and biomass burning volatile organic compound emissions on regional and seasonal variations in secondary organic aerosol"

In the introduction (section 1), we have also highlighted the overall aim of the manuscript, to include the major VOC emissions sources of SOA:
page 6 lines 10-13 of updated manuscript:
"In this study, a global chemistry and aerosol model (UKCA) is used to simulate SOA concentrations from all the VOC emission source types described above: monoterpene, isoprene, anthropogenic and biomass burning. Other mechanisms of SOA production, such as S/IVOCs and

heterogeneous production, may also be important, but in this study, the focus is on VOCs."

6. *Page 16: Welgegund: Can biomass burning be a large missing SOA source at this cite rather than anthropogenic SOA? For example, Shrivastava et al. 2015 report that including biomass burning as SOA source improves model-measurement agreement in terms of seasonality of SOA at this site.*

Yes, we agree that biomass burning could be the missing source. Text has been added to section 4.3.2 to make this point.
Page 21 lines 1-2 of updated manuscript:
"In contrast, Shrivastava et al. (2105) find good agreement between simulated and observed OA when considering ARCTAS measurements, primarily to their simulation of biomass burning SOA."

7. *Figure 14: The various lines need captions. The text and the legends in this figure need to be made clearer for readability. For example, it's not clear which is monoterpene only versus total all source SOA simulation.*

Apologies for not noticing the legend was missing from this figure. Figure 14 has been updated.

8. *The authors report that inclusion of biomass burning SOA source does not improve model performance with respect to aircraft measurements especially during ARCTAS. In sharp contrast, Shrivastava et al. 2015 reported a large increase in model performance especially at high altitudes, with respect to ARCTAS field campaign when they included biomass burning SOA source. Clearly, this reflects the large difference in biomass burning SOA treatment between this study and Shrivastava et al. 2015 study. Some discussions about why the authors don't see an improvement due to biomass burning SOA is warranted and also how their SOA treatment for biomass differs from Shrivastava et al. 2015 study*

Within the original manuscript, we have attributed the lack of improvement between modelled and observed OA when $VOC_{BB}$ is included to two factors: 1) the global SOA production rate from $VOC_{BB}$ is relatively small in comparison to other sources, and 2) peak biomass burning emissions (e.g. South America and Africa) are in different regions to the biomass burning aircraft campaigns (e.g. North America) - page 15 line 18 and page 18 line 5, page 19 line 24 of original manuscript. In addition to the new text added in response to point 6 above, we have added further discussion of the difference in biomass burning SOA treatment between this study and Shrivastava et al. (2015):

Page 21 lines 3-7:

"These results highlight that biomass burning remains a highly uncertain source of SOA. Here, biomass burning SOA is considered from VOCs, with a global annual-total emission rate of 49 Tg $a^{-1}$, and injected at the surface. Contrastingly, Shrivastava et al. (2015) treated biomass burning SOA from S/IVOCs, with global annual-total emissions of 450 Tg $a^{-1}$, which are injected at the surface as well as higher altitudes. Clearly, further research is required to identity the dominant sources of biomass burning SOA, as well emissions estimates and chemistry."

**Referee #2**

*Kelly et al. update a global chemistry-climate model (UKCA) for new sources of secondary organic aerosol (SOA) and report on model predictions and the model-measurement comparison from this updated model. They find that, on average, the inclusion of new sources of SOA improves model performance against organic aerosol (OA) mass concentrations, POA-SOA splits, and OA vertical profiles but caution that the model still does not include some major SOA formation pathways and processes (e.g., varying volatility, aqueous chemistry) and remains unconstrained in the southern hemisphere due to a paucity of observations.*

*Kelly et al. have done an excellent job of reporting results from the model simulations and the model-measurement comparison. The manuscript is also very well written and makes for very easy reading. The findings and discussion from this work will be helpful to the community. However, I found that the methods used did not reflect the gaps/uncertainties discussed in the introduction nor come close to the current state-of-the-science for treating SOA formation pathways and processes in atmospheric models. While novel for the UKCA, the sources/formation pathways explored in this work are routine for some of the other global chemistry-climate models and I was struggling to see how this work was novel and offered fresh insights into the SOA budget that haven't been explored in earlier work. This work definitely needs to be published but I am not comfortable recommending publication in Atmospheric Chemistry and Physics given the lack of novelty.*

We thank the reviewer for their insightful comments. We fully agree that the treatment of SOA within this composition-climate model is not state-of-art. New SOA formation pathways, such as aqueous phase production and the conversion from POA, have recently been identified as potentially important sources of SOA. Also, complex treatments of SOA, such as the volatility basis set (VBS), are currently being developed. However, in a recent review paper on SOA modelling, it appears that these recent advances in SOA understanding have are not fully accounted for in today's global models, especially composition-climate models (http://dx.doi.org/10.1007/s40641-018-0092-3). This is demonstrated by the results from AEROCOM. In this study, of the 31 state-of-the-art chemistry climate models and general circulation models included, only 12 treat OA as semi-volatile, only 3 include aqueous production of SOA, and only 1 uses the volatility-basis set (VBS) (Tsigaridis et al., 2014). Therefore, we feel our research is relevant to many other researchers that are developing their SOA schemes in their respective models, and we highlight that there are still key uncertainties associated SOA production from VOCs. We do realise that we have not been clear about the focus of our study which is to gain insights into the different VOC emission sources that lead to SOA production which may have led to misinterpretation of our objective. Hence as discussed above in our response to reviewer 1 (comment 5) we have clarified the overall aim of our paper in the introduction and modified our title to:
"The impact of biogenic, anthropogenic and biomass burning volatile organic compound emissions on regional and seasonal variations in secondary organic aerosol"

*I have listed just a few of my concerns related to the methods:*

1. *The POA (e.g., Robinson et al. (2007, 2010)) and SOA (e.g., Donahue et al., 2012) need to be treated as semi-volatile and reactive to better model the OA mass concentrations and its sensitivity to chemistry and changes in temperature. One example where this would influence one of the findings reported in this work is that SOA monolayers coated on POA could evaporate with dilution/chemistry and transition POA back into its hydrophobic mode. Another complication related to treating the semi-volatile nature of OA that has surfaced recently is if organic particles achieves instantaneous equilibrium with the organic vapors and how the phase state (i.e., diffusion limitations within the particle) might influence the timescales to achieve equilibrium (e.g., Shiraiwa et al., 2017). A semi-volatile treatment should be trivial to include with say a 2-product model (e.g., Odum et al., 1997).*

- This is a very interesting point, the volatility of OA is an extremely important factor in governing SOA. However, within the literature, there is no consensus over the volatility of OA. Evidence for both semi-volatile (Robinson et al., 2007;Donahue et al., 2012), and non-volatile (Jimenez et al., 2009;Cappa and Jimenez, 2010) behaviour of OA exists. Of the 31 state-of-the-art global chemistry transport models (CTMs) and general circulation models (GCMs) included in AEROCOM, only 12 treat OA as semi-volatile (Tsigaridis et al., 2014). However, we do note that the volatility will have substantial impacts on OA, as demonstrated in a global modelling study by Shrivastava et al. (2015). However, we do not feel that that this study needs a 2-product approach. As discussed above, the objective of our study is to investigate the role of different VOC emissions on the SOA budget and model agreement with observations. Hence, to investigate these different VOC source contributions we need to maintain an identical volatility treatment across all simulations. However, we had added additional text to our introduction to highlight the uncertainty over the volatility of OA and included the citations provided. Also, we thank the reviewer for the references with regards to particle viscosity and this has also been added to the introduction.

  Page 5 line 25-30 of updated manuscript;
  "Volatility is also another important aspect of OA, with substantial impacts on SOA production. There is evidence for both semi-volatile (Robinson et al., 2007;Donahue et al., 2012), and non-volatile (Jimenez et al., 2009;Cappa and Jimenez, 2010) behaviour of OA. Of the 31 state-of-the-art global models included in AeroCom, only 12 treat OA as semi-volatile (Tsigaridis et al., 2014). Recently, the effects of volatility on SOA were quantified in a global modelling study by Shrivastava et al. (2015). These authors estimate that the global annual-total SOA production rate varies by almost a factor of 2 depending on whether OA is treated as semi-volatile or non-volatile. The particle-phase state may be another important factor but is poorly characterised (Shiraiwa et al., 2017), and is dependent on relative humidity and SOA precursor (Hinks et al., 2016;Bateman et al., 2016)"

2. *There is plenty of evidence that the chemical lifetimes and SOA mass yields are very different for different SOA precursors (in addition to being a function of the OA mass loading), which can be very easily reflected in this work (regardless of whether the OA is treated as semi-volatile or non-volatile). Furthermore, SOA mass yields need to be corrected for vapor wall loss artifacts experienced in chamber experiments (e.g., Zhang et al., 2014).*

- This is a good point. Firstly, we agree that the chemical lifetime of SOA precursors (i.e. the lifetime of VOCs with respect to oxidation) will vary from one VOC species to another. This difference in reactivity is explicitly accounted for in the cases of isoprene and monoterpene, as they have laboratory-constrained estimates of reaction kinetics (as shown in Table 2). However, $VOC_{ANT}$ and $VOC_{BB}$ are not specific compounds as they are surrogate species representing anthropogenic VOC and biomass burning VOC respectively. Hence, these two lumped compounds reflect a mixture of VOC compounds. As we do not know the chemical speciation of this mix, we cannot apply a molecular-specific rate constant for these species. In light of the uncertainties over the reactivity of $VOC_{BB}$ (also highlighted by reviewer 1 comment 4), we have performed an additional simulation to test the sensitivity of SOA to assumed reactivity and have including these results in section 4.2. Interestingly, we find little impact on the global SOA production rate. In addition, we do not feel that there is clear consensus over the SOA yields for various precursors. For known SOA precursors, reaction yields vary substantially from one study to another. Furthermore, $VOC_{ANT}$ and $VOC_{BB}$ are unknown species, hence, again, we cannot select molecular-specific reaction yields from laboratory studies. Even monoterpene is a mixture of compounds with SOA yields varying substantially (Zhao et al., 2015). Also, we have chosen to implement the new sources of SOA with identical reaction yields initially as this allows the difference in SOA production from the various VOC sources to be attributed to differences in emissions. We acknowledge that reaction yields may vary from one compound to another. However, we did explore the potential for differences in reaction yields within the study in the Iso(3%) and Ant(40%) simulations. To address this concern we have added text to the manuscript to 1) test the sensitivity of SOA to assumed reaction kinetics (section 4.2), 2) provide more evidence for the uncertainty in reaction yields including wall loss effects (section 1), and 3) explain why we selected identical reaction yields for the initial simulations (section 2.1).

   Page 14 line 43 – page 15 line 8:
   "The influence of assumed reactivity on simulated SOA from biomass burning was investigated in an additional sensitivity simulation where $VOC_{BB}$ adopts the reactivity of naphthalene (Table 2; section 2.3) , an aromatic species which has been used as a surrogate compound for IVOCs (Pye and Seinfeld, 2010). Compared to monoterpene, naphthalene is roughly 50 % less reactive

(Atkinson and Arey, 2003). However, despite this substantial reduction in reactivity, the global annual-total SOA production rate from biomass burning VOCs is reduced by less than 1 %. Also, the simulated spatial distributions are almost identical for the two $VOC_{BB}$ species. Like all other SOA precursors in this study, $VOC_{BB}$ does not undergo dry or wet deposition. Therefore, a reduction in the reactivity of $VOC_{BB}$ does not affect the fate of this compound."

- Page 5 line 10-19:
"The SOA reaction yields, which are typically determined in environmental chambers, are highly uncertain. Firstly, the VOC sources implemented into models often represent a mixture of different compounds, which may have differing reaction yields. For example, a recent environmental chamber study found that the SOA reaction yield varies from 2.5 to 16.9 % depending on the particular monoterpene species (Zhao et al., 2015). Secondly, reaction yields vary substantially from one study to another. Considering aromatic compounds, which are typically associated with SOA production from anthropogenic and biomass burning, estimates of SOA reaction yields from aromatic compounds from different chamber studies range from 1 to 73 (Odum et al., 1996;Odum et al., 1997;Song et al., 2005;Ng et al., 2007b;Chan et al., 2009). Part of this uncertainty is due to differences in environmental chamber conditions. For aromatic compounds, factors such as relative humidity (Hinks et al., 2018) and $NO_x$ (Ng et al., 2007a) influence oxidation pathways which, in turn, control the yields of SOA. Finally, within an environmental chamber, uptake of organic compounds by the surface walls (known as 'wall losses') can occur. Traditionally, this process was assumed to be non-negligible, resulting in the potential for environmental chamber studies to underestimate the reaction yield. Zhang et al. (2014) found that reaction yields of toluene, an anthropogenic source of SOA, were under estimated by a factor of four due to wall losses during chamber studies. This has a significant effect on simulated SOA production. Updating reaction yields to account for wall losses in global models resulted in an increase in the global annual biogenic SOA production rate from 21.5 to 97.5 Tg (SOA) $a^{-1}$ (Hodzic et al., 2016)."

Page 8 lines 22-28:
"As discussed in Section 1, reaction yields vary from one study to another, as well as within individual studies. Furthermore, $VOC_{ANT}$ and $VOC_{BB}$ are surrogate compounds, representing a mixture of species and, therefore, preventing selection of molecule-specific reaction yields from laboratory studies. The initial assumption of identical reaction yields for all species does not negate the findings from laboratory studies, which suggest reaction yields are highly dependent on both molecular structure. However, the substantial uncertainties of reaction yields, coupled with these species representing lumped species, prevents accurate selection of laboratory-derived reaction yields for specific compounds."

3. *Emissions of semi-volatile and intermediate-volatility organic compounds contribute significantly to anthropogenic SOA precursors (and possibly even biogenic SOA precursors) and need to be explicitly modeled (e.g., Jathar et al., 2014). These might help reduce the under predictions in urban areas and elsewhere. On a related note, it is unclear to me how the model-measurement comparison at urban locations needs to be evaluated. What fraction of the under-prediction can be seemingly attributed to the model resolution?*

- We certainly agree that S/IVOC emissions may improve model under predictions of observed OA. The potential importance of S/IVOCs were highlighted in the introduction (Page 4 Line 6 of original manuscript). Also, in our model-to-measurement comparison text in section 4.3.1, we highlighted that our model negative bias may be improved by inclusion of S/IVOCs (Page 14 Line 26 of original manuscript). However, we do not feel that S/IVOC emissions need to be considered in this study as our overall aim is to quantify the role of different VOC emissions on the global SOA budget and model agreement with observations. Ad discussed above, we have changed the title of the manuscript to be clear about the purpose of our study.

- We feel that model-to-measurement comparison needs to be evaluated consistently across all environment sites. We used a suite of statistical tools to evaluate model performance at all sites, then repeated the process for the various site types. Generally, we found that a negative model bias was systematic across urban, urban downwind and remote, indicating missing sources of SOA. However, we noted that the model negative bias in urban environments is probably due to both missing sources and the coarse model resolution (Page 14 Line 20 of original manuscript). To establish the impacts of model resolution on model agreement with observations at urban locations would require further model simulations with varying degrees of resolution. We feel this is outside the scope of this study. Alternatively, some global modelling studies chose to evaluate model data against remote and urban downwind sites only (Hodzic et al., 2016) . However, SOA is important from an air quality perspective. Therefore, we feel including this model-measurement comparison at urban locations and highlighting this negative bias in urban regions that is not resolved with the addition of $VOC_{ANT}$ is important.

4. *Globally, aqueous processing of organic compounds in aerosol water and clouds is probably a very important source/sink of OA and needs to be included (McNeill et al., 2015). At the very least, one needs to consider IEPOX, glyoxal, and in-cloud formation of a few dominant organic acids in the model.*

- We agree, aqueous production of SOA may be important for SOA, and was highlighted in the introduction and conclusions (Page 3 Line 7 and Page 20 Line 6) in the original manuscript. But as discussed above aqueous production does not fall within the objective of our study, which is to quantify

the role of VOC emission on SOA and we have modified the manuscript title and overall aim for clarity.

- Also, aqueous phase production of SOA is uncertain and in the recent AEROCOM study by Tsigaridis et al. (2014) only 3 of the 31 models include this process. Therefore, we have not included aqueous production of SOA, but we have expanded the discussion of this source within our introduction.

Page 4 line 17-25:

[revised manuscript text omitted]

---

## Author Response (AR2)

**We thank the reviewer for their reports. For each reviewer's comment (reiterated in italics), we have provided a response, and we have also provided the modified text within the updated manuscript (indicated by quotation marks). In our revised manuscript, modified text is highlighted using tracked changes.**

**Report #1**

- We thank the reviewer and are happy that they are pleased with the revised manuscript.

**Report #2**

*I generally agree with the author's stance on most of my comments and I think the change in the framing of the problem – one of focusing on the SOA formation from VOCs – responds to my major concern with this manuscript. There are however a few things I would like to point out from the response that caught my attention.*

1. *For the response on comment #1, I don't agree with the simple explanation added to the manuscript to justify the use of a non-volatile POA that 'there is evidence for both semi-volatile and non-volatile OA'. The issue of volatility is nuanced and should be discussed accordingly. For instance, source measurements of combustion POA performed at different dilutions (e.g., May et al., 2013a,b,c) and thermodenuder measurements of laboratory-generated (e.g., Huffman et al., 2009a) and field OA (e.g., Huffman et al., 2009b) clearly demonstrate that OA is semi-volatile with some fraction of non-volatile material. However, certain experiments with model SOA systems exhibit delays in evaporation that could be a result of diffusion-linked limitations to evaporation (e.g., Vaden et al., 2009). Simply stating that there is evidence for both and then treating the POA as non-volatile suggests that the weight of the literature points towards POA being more non-volatile, which is definitely not the case.*

- We have now revised our text on OA volatility in light of this discussion, which we thank the reviewer for. We have revised the text on volatility in the introduction (Section 1), and re-iterated how the treatment of OA as non-volatile in UKCA, contrary to some field and lab studies in the methods (Section 2.1)

- Page 6 lines 3-13
  "Volatility is also another important aspect of OA. Whereas oxidation products from biogenic VOCs are predominantly compounds with extremely low volatility, so-called ELVOCs, (e.g. Ehn et al., 2014), there is considerable evidence that anthropogenic SOA is primarily composed of semi-volatile material. A large component of POA is also semi-volatile. For example, combustion POA, either generated from gasoline (May et al., 2013b) or diesel (May et al., 2013c) within laboratory studies suggests POA is primarily semivolatile (i.e. only a small proportion consists of ELVOCs) . Also, analysis of biomass burning POA emitted from the burning of vegetation within laboratory studies suggests POA is semi-volatile (May et al., 2013a). Furthermore, analysis of OA, generated in both laboratories (Huffman et al., 2009b) and field campaigns (Huffman et al., 2009a) suggests OA is semi-volatile. However, some specific cases suggest a proportion of OA is non-volatile (Cappa and Jimenez, 2010;Jimenez et al., 2009;Vaden et al., 2011). Considering laboratory and field studies suggest OA is dominated by the semi-volatile fraction, the treatment of volatility of OA within global models ranges substantially."

- Page 8 line 12 - 15
"In this study, as with the majority of global aerosol models (e.g. Tsigaridis et al., 2014), OA is treated as a non-volatile species, the emission or chemical yield implicitly reflecting only the particle phase component of the OA.  As we discuss in section 1, anthropogenic OA in particular has a substantial semi-volatile component, which will introduce an important mode of variability to aerosol properties in industrialised regions.

2. *The response in comment #2 makes it sound that we do not know a lot about the differences in VOC reaction rates, SOA mass yields, and speciation of anthropogenic and biomass burning emissions. I would argue that a lot has been understood over the past decade (e.g., motor vehicle VOC = May et al., 2014; Zhao et al., 2015; Zhao et al., 2016, biomass burning = Stockwell et al., 2015; Hatch et al., 2017) but that there is still a lot more to learn. Also, I disagree with some of the inferences made from the literature. For instance, the differences in SOA mass yields between different aromatic compounds can be explained if the aromatics are split into low and high yield categories (Odum et al., 1996). NOx effects on SOA formation from aromatics can be modeled through a competition of the RO2 radical with NO and HO2 (Henze et al., 2008). Furthermore, for a model such as that used in this work adding model species can be computationally expensive and need to be kept to a minimum. However, their reactivity and SOA mass yield could be informed by a weighted average of the mixture of species used to represent the model species.*

- We thank the reviewer for this comment. Upon reflection, we realise that our discussion on VOC speciation and SOA yields is not reflective of the recent advances in VOC speciation and SOA yields. Also, we have begun work modifying UKCA to incorporate NOx-dependent yields, which we hope to publish in a separate paper. We have chosen to revise our discussion on SOA yields (Introduction; Section 1) and anthropogenic and biomass burning VOC speciation (Methods; Section 2.2).

- Page 5 lines 11-27

[revised manuscript text omitted]